# Secreted ISG15 induced by *Chlamydia trachomatis* infection exerts immunomodulatory effects on IFN-γ defense and inflammation

Yongxia Guo[1,2], Sigrun V. Stulz[3], David Komla Kessie[1], Nadine Vollmuth[1¤], Tommaso Torcellan[3], Klaus-Peter Knobeloch[4], Georg Gasteiger[3], Thomas Rudel[1]*

**1** Department of Microbiology, Biocenter, Julius-Maximilians-Universität Würzburg, Würzburg, Germany, **2** College of Veterinary Medicine, China Agricultural University, Beijing, China, **3** Würzburg Institute of Systems Immunology, Max Planck Research Group at the Julius-Maximilians-Universität Würzburg; Würzburg, Germany, **4** Faculty of Medicine, Institute of Neuropathology, University of Freiburg, Freiburg, Germany

¤ Current affiliation: Department of Biological Sciences, University of Alabama, Tuscaloosa, Alabama, United States of America

* thomas.rudel@uni-wuerzburg.de

## Abstract

Interferon-γ (IFN-γ) is an immunoregulatory cytokine essential for cellular immunity against intracellular pathogens, including *Chlamydia*. Interferon-stimulated gene (ISG) 15, a member of the ubiquitin family, contributes to host resistance to viral and bacterial infections. ISG15 can exist either in an unconjugated form or covalently attached to host proteins through a process known as ISGylation. Here, we show that infection with *Chlamydia trachomatis* (Ct) induces the expression and secretion of ISG15 in human primary cells and mouse female genital tract (FGT) organoids. ISG15 secretion by genital tract epithelial cells resulted in increased IFN-γ release from natural killer (NK) cells. The production of IFN-γ by NK cells in response to ISG15 was completely abolished in NK cells lacking the interleukin-18 receptor alpha (IL-18Ra), demonstrating a co-stimulatory effect of ISG15 with IL-18 in enhancing IFN-γ release. ISG15 was secreted into the FGT and was involved in controlling bacterial load in a murine infection model. Furthermore, ISG15 reduced macrophage responsiveness to IFN-γ as an M1-polarizing signal for pro-inflammatory responses, potentially "shielding" macrophages from excessive IFN-γ. Evidence of uterine horn pathology and reduced IL-10 levels in the FGT of infected ISG15⁻/⁻ mice further supports a critical dual function of ISG15 in controlling chlamydial infection and modulating the resulting inflammatory responses.

## Author summary

In our research, we explored how the immune system responds to *Chlamydia trachomatis* infection, focusing on a molecule called ISG15. This molecule is

**Data availability statement:** All relevant data are within the manuscript and its Supporting Information files.

**Funding:** The work was supported by the DFG funding SFB1583 (DECIDE; Project B01 to G.G. and Project A05 to T.R.) and the European Research Council (ERC) under the European Union's Horizon 2020 research and innovation program (ERC-2018-ADG/NCI-CAD) to T.R.. The funders had no role in study design, data collection and analysis, decision to publish, or preparation of the manuscript.

**Competing interests:** The authors have declared that no competing interests exist.

typically activated by interferon-gamma (IFN-γ), a key player in defending the body against infections. We found that ISG15 not only increases during infection but is also secreted by cells in the female genital tract. Interestingly, this secreted ISG15 helps natural killer (NK) cells produce more IFN-γ, but only if another molecule, IL-18, is also involved. This shows ISG15 and IL-18 work together to boost immune defense. At the same time, ISG15 appears to protect the body from going overboard - by dampening how macrophages (another immune cell type) respond to IFN-γ, it prevents excessive inflammation. In mice that lacked ISG15, there were signs of more severe infection and tissue damage, along with less of the anti-inflammatory molecule IL-10. Overall, our findings reveal a balancing act: ISG15 helps fight infection while also keeping inflammation in check, highlighting its dual role in immune defense and regulation.

## Introduction

*Chlamydia trachomatis* (Ct) is a bacterial pathogen that causes blinding trachoma and represents the most common sexually transmitted disease (STD) in humans worldwide [1,2]. In urogenital infections, Ct infects cervical epithelial cells in women and urethral epithelial cells in men. Because asymptomatic infections are common [3] and therefore go untreated, *Chlamydia* can ascend to the fallopian tubes and cause serious conditions such as pelvic inflammatory disease (PID), tubal factor infertility (TFI), and ectopic pregnancy [4,5].

As an obligate intracellular bacterium, Ct uses epithelial cells of the genital tract to replicate in a biphasic developmental cycle. Elementary bodies (EBs), the infectious, non-replicative form, enter the cell and transform into the replicative reticulate bodies (RBs). After several rounds of multiplication, the RB transform back into EB and leave the cell by extrusion or after cell lysis to start the next cycle of infection and replication [6]. In addition to active infection, *Chlamydia* can establish a persistent infectious state under stress conditions such as exposure to antibiotics or interferon-γ (IFN-γ) [7,8]. During persistent infection, *Chlamydia* is capable of maintaining a prolonged presence within its host cell, which can lead to chronic infection and the development of severe pathology [9,10].

The immune response directed against the Ct-infected genital tract is complex. Intracellular pathogen recognition receptors in epithelial cells detect the bacteria and initiate signaling cascades leading to the secretion of cytokines and chemokines [11,12]. As a result, innate immune cells, including neutrophils, natural killer (NK) cells, dendritic cells (DCs) and macrophages (Mφ), etc., infiltrate the infected site to orchestrate the local immune response and to limit the infection [12,13]. These innate immune cells secrete numerous cytokines and chemokines that can ultimately cause excessive inflammation. Tissue damage and chronic inflammation caused by innate immune cells have been implicated in tubal pathology and PID severity in chlamydial genital tract infections [14,15]. A recent study showed that activation of submucosal macrophages can directly lead to sterile inflammation and tissue damage in

the female genital tract (FGT) [16]. In response to infection, macrophages orchestrate an early-stage pro-inflammatory response by secreting a variety of inflammatory mediators, such as TNF-α, IL-1, IL-6, IL-12, and nitric oxide. These mediators further recruit, activate, and expand antimicrobial immune cells, thereby orchestrating the immune response to infection. This pro-inflammatory response is initially beneficial because it facilitates the defense against the pathogen. However, if not promptly controlled, it can cause substantial collateral tissue damage. To counteract this tissue damage, subsets of macrophages acquire an anti-inflammatory and reparative functional state, aimed at restoring tissue homeostasis [17,18]. A dysregulation of macrophage polarization and functions may contribute to persisting chronic inflammation and pathology [19]. Nevertheless, the mechanisms regulating local immunity, inflammation and pathology in the genital tract in response to Ct-infection remain incompletely understood.

IFN-γ-mediated depletion of tryptophan and extensive shutdown of the host metabolism in human epithelial cells have been demonstrated to inhibit chlamydial growth and could eradicate infection under continuous IFN-γ exposure. This mechanism has been identified as a critical factor in the clearance of chlamydial infection and the protection against reinfection in the FGT [20–22]. These findings are consistent with those of animal studies and patients, which indicate that Ct-specific IFN-γ–producing CD4+ T cells are strongly associated with protective responses and are therefore considered to be the primary source of IFN-γ [22–24]. However, CD4+ T cell-dependent IFN-γ protection driven by the adaptive immunity cannot account for protection during the early stages of infection because naïve *Chlamydia*-specific CD4+ T cells must first be primed in the lymph nodes, before they can migrate as activated and differentiated T cells to the genital mucosa and secrete IFN-γ [25]. Early IFN-γ production by innate immune cells is therefore essential for systemic control of *Chlamydia* in mice, whereas IFN-γ from CD4+ T cells appears redundant for preventing bacterial dissemination in the FGT [26].

NK cells are a significant cellular source of IFN-γ [27]. NK cells are present throughout the FGT as part of the mucosal immune system [12]. They respond to a wide range of stimuli, including cytokines produced by infected and myeloid cells (most prominently type I IFN, IL-12 and IL-18) as well as signals indicating cellular stress, infection, or oncogenic transformation. In particular, NK cells play a critical role in limiting *Chlamydia* infection by producing IFN-γ early in the genital tract infection, and by contributing to the Th1 immune response [28,29]. IL-18, also known as interferon-gamma inducing factor, is primarily derived from myeloid cells but is also emerging as an epithelium-derived cytokine that contributes to the host defense against intestinal infections [30,31]. However, the regulatory mechanisms governing IFN-γ responses within the FGT during *Chlamydia* infection remain poorly understood.

*Chlamydia* infection in epithelial cells is sensed by the cGAS-STING signaling pathway, which induces type I interferons as a major mediator of the anti-chlamydial immune response [32,33]. ISG15 (IFN-stimulated gene 15) is a 15-kDa member of the ISG family that is highly induced by type I interferons (IFNs) [34]. ISG15 has been demonstrated to function in three distinct forms. Firstly, it acts as a ubiquitin-like modifier (Ubl), covalently conjugated to target proteins via the process of ISGylation [35,36]. Secondly, it exists as an unconjugated intracellular protein, which has been shown to negatively modulate the type I IFN (IFN-α/-β) signaling pathway in human cells [37]. Thirdly, a secreted form of ISG15 has been shown to exert immunomodulatory effects on NK cells, neutrophils and DCs [38,39], and acts as an inducer of IFN-γ and IL-10 production in human NK cells and lymphocytes. ISG15 can be secreted by neutrophils, lymphocytes, monocytes, plasma cells, and intestinal and bronchial epithelial cells [40–42]. The function of ISG15 during viral infection has been extensively studied [34], but its role in bacterial infection, particularly intracellular bacteria, is less clear. Humans with inherited ISG15 deficiency are more susceptible to mycobacterial infection, as a result of impaired IFN-γ immunity [40]. These patients lack extracellular ISG15, which functions in synergy with IL-12 to enhance IFN-γ secretion from T and NK cells [40]. ISGylation, the covalent modification of proteins with ISG15, also protects mice against *M. tuberculosis* infection [43]. During *Listeria monocytogenes* infection, the expression of ISG15 is induced and ISG15 counteracts bacterial replication in an ISGylation-dependent manner both *in vitro* and *in vivo* [44]. The transcription of ISG15 in epithelial cells is strongly upregulated by Ct infection [45,46], but the effects of ISG15 in the infected FGT are currently unknown.

Here we show that Ct-infection induces the secretion of ISG15 from primary host cells. We investigated the function of secreted ISG15 in the innate immune response against Ct using murine FGT organoid and murine *in vivo* infection models. Our findings demonstrate a pivotal role for secreted ISG15 in the regulation of the IFN-γ-dependent defense against *Chlamydia* infection. Additionally, we demonstrate that ISG15 regulates the progression of chlamydial genital disease by promoting an M2-like phenotype (CD163+) in macrophages and inhibiting their proinflammatory polarization in response to IFN-γ, thereby mitigating excessive inflammatory responses.

## Results

### Infection with Ct induces expression and secretion of ISG15

To investigate whether ISG15 is induced by Ct infection, we infected different human cell lines and primary cells or treated them with IFN-α, a known stimulator of ISG15 production. ISG15 was barely detectable in Ct infected HeLa 229 cells by Western blot but highly induced by IFN-α (Fig 1a). To investigate the ISG15 response in primary cells, we examined ISG15 expression in primary cells from human ovarian fimbriae (referred to as Fimb cells), in human umbilical vein endothelial cells (HUVEC cells), and in primary ectocervical epithelial cells isolated from biopsies (EcCxEp cells) infected with Ct. All infected primary cells produced high levels of ISG15 as shown by Western blotting (Fig 1a). Since extracellular ISG15 has cytokine-like activities [40,41], we investigated whether ISG15 could be released from Ct-infected cells. To assess ISG15 secretion, cell-free supernatants of HUVEC, Fimb, and HeLa 229 cells infected with Ct and the respective controls were collected and analyzed by ELISA. Ct infection significantly increased the release of ISG15 in both HUVEC and Fimb cells, but not in HeLa 229 cells (Fig 1b and S1 Data). It is noteworthy that, in contrast to infection, cells stimulated with IFN-α exhibited only a mild, non-significant release of ISG15. This suggests that the robust secretion of ISG15 in primary host cells results from infection, rather than cytokine stimulation.

Since secreted ISG15 has been shown to have immunoregulatory functions, we continued our study in mice with the aim to investigate the role of ISG15 in a fully immunocompetent infection model. To initially assess the expression of ISG15 in the primary target cells for *Chlamydia*, namely the epithelial cells, an organoid model of the mouse female genital tract (mFGT organoids) was established as a cellular infection model. Tissue from the mFGT was processed into a single-cell suspension containing stem cells with the capacity to generate a range of female genital tract organoids [47]. These cells were cultured in a 3D Matrigel matrix containing Wnt-3A, R-spondin-3, and noggin (Fig 1c; see Materials and Methods for details). Infection of these mFGT organoids with Ct for 3 and 5 days induced high ISG15 expression (Fig 1d), demonstrating the production of ISG15 in mFGT tissue.

We observed that most Ct inclusions were released apically into the organoid lumen, and only a few inclusions were released basolaterally (S1a Fig). ISG15 secretion was investigated in this infection model in intact organoids and in organoids that were disrupted by up-and-down pipetting prior to harvesting the supernatant for ELISA. Significant release of ISG15 was observed only in Ct-infected organoids and not in uninfected controls in both disrupted and intact organoids (Fig 1e), demonstrating that mFGT organoids express and secrete ISG15 in response to Ct infection. To determine whether ISG15 induction is conserved across *Chlamydia* strains, we infected mFGT organoids with *Chlamydia muridarum* (Cm), which showed robust ISG15 expression (S1b Fig) and secretion (Fig 1f). These results confirm that both Cm and Ct trigger ISG15 production in a host-relevant model.

### Immunoregulatory effects of extracellular ISG15 on murine immune cells

Extracellular ISG15 has been demonstrated to enhance IL-12-induced IFN-γ production and to induce IL-10 secretion from human peripheral blood mononuclear cells (PBMCs) [40,41,48], which are critical cytokines for Ct clearance and resolution of inflammation, respectively. Therefore, we investigated IFN-γ secretion in murine immune cells following stimulation with recombinant ISG15 and recombinant IL-12. ISG15 significantly enhanced IFN-γ release in murine splenocytes in synergy with IL-12 (Fig 2a). NK cells are known to be the predominant cell type within human PBMC populations

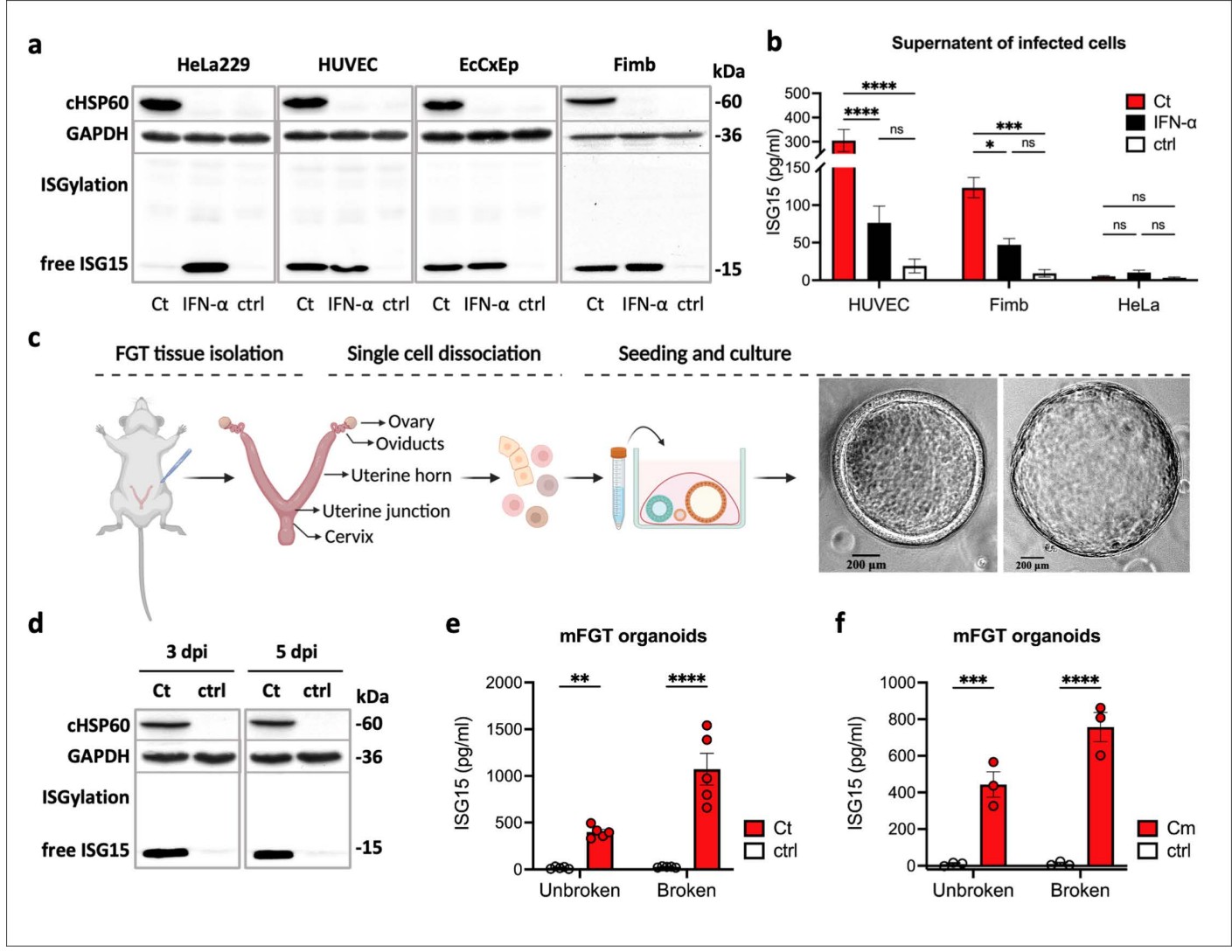

**Fig 1. ISG15 expression and secretion is induced in human primary cells and mouse female genital tract organoids upon Ct infection.** a. ISG15 expression in human cells infected with Ct. HUVEC, Fimb, EcCxEp, and HeLa 229 cells were either left uninfected, infected with Ct (MOI 1), or stimulated with human IFN-α (100 ng/ml) for 36h. ISG15, HSP60 (chlamydial infection), and GAPDH (loading control) expression in cell lysates were analyzed by Western blotting. A representative blot from three independent experiments is shown. b. Secretion of ISG15 in Ct-infected human cells. Cells were either left uninfected, infected with Ct (MOI 2), or treated with 100ng/ml human IFN-α for 36 h. Secreted ISG15 in cell-free supernatants was quantified by ELISA. Data were analyzed by two-way ANOVA followed by Tukey's test, n = 3. c. Schematic of mouse female genital tract (mFGT) organoids generation and representative bright-field images of different mFGT organoids generated 2-3 weeks after seeding. Single cells from mFGT tissue were cultured in Matrigel to form 3D organoids. d. ISG15 expression in mFGT organoids infected with Ct for 3 and 5 days. ISG15, HSP60, and GAPDH expression of infected and non-infected mFGT organoids were analyzed by Western blotting. A representative blot from three independent experiments is shown. e. ISG15 secretion by mFGT organoids infected with Ct for 5 days. Organoids were either intact or disintegrated (to release luminal contents). ISG15 levels in cell-free supernatants were analyzed by ELISA. Data were analyzed by Mann-Whitney test, n = 5. f. ISG15 secretion by mFGT organoids infected with Cm for 5 days, analyzed as in (e). Mann-Whitney test, n = 5. (Statistical significance is indicated as follows: *p ≤ 0.05; **p ≤ 0.01; ***p ≤ 0.001; ****p ≤ 0.0001; ns, not statistically significant). Fig 1c was created in BioRender (2025; https://BioRender.com/3ifquqa).

that produce IFN-γ in response to co-stimulation by ISG15 and IL-12 [41]. Therefore, we enriched primary NK cells from murine splenocytes to a purity between 85%-96% by negative magnetic separation (S2 Fig). ISG15 enhanced IFN-γ release in these murine NK cells in synergy with IL-12 (Fig 2b). However, the effect of ISG15 was less pronounced in

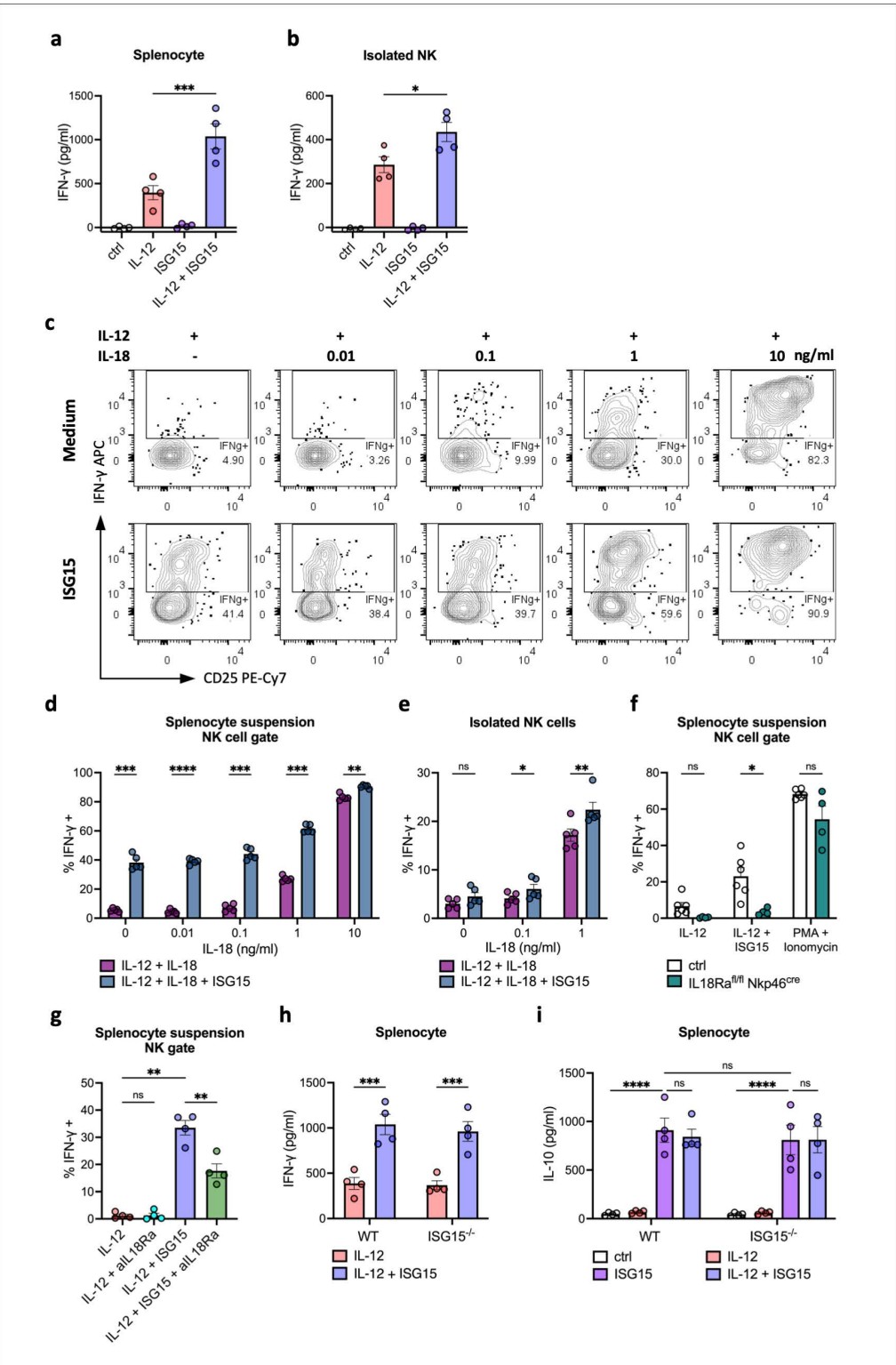

**Fig 2. Extracellular ISG15 stimulates mouse immune cells to produce IFN-γ and IL-10.** a. ISG15 in synergy with IL-12 induces IFN-γ secretion from mouse splenocytes. Splenocytes were either left untreated or stimulated with IL-12 (20 ng/ml), ISG15 (250 ng/ml), or a combination of IL-12 and ISG15 for 36 h. IFN-γ production in the supernatant was measured by ELISA. Data were analyzed using one-way ANOVA with Tukey's test, n = 4. b.

ISG15 with IL-12 induces IFN-γ secretion from mouse NK cells. Isolated NK cells were stimulated and analyzed for IFN-γ levels as described in (a), n = 4. c. ISG15 enhances IFN-γ production by NK cells in splenocyte suspensions. Splenocytes were stimulated for 6h in medium containing IL-12 (20 ng/ml) and IL-18 (as indicated), with or without ISG15 (250 ng/mL). Representative FACS plots are shown. n = 5, representative of two independent experiments. d. The percentage of IFN-γ + NK cells in splenocyte suspensions quantified from FACS assay in (c). Data were analyzed using two-way ANOVA with Šídák's test, n = 5. e. Isolated NK cells show a low ISG15 effect. Isolated NK cells (purity > 90%) were stimulated and analyzed for the ratio of IFN-γ + NK cells as indicated in (d), n = 5, representative of two independent experiments. f. NK cells require IL-18 receptor to respond to ISG15 stimulation. Splenocytes from IL18Ra^fl/fl NKp46^cre or control mice (WT or NKp46^cre) were stimulated at $10^6$ cells/ml (IL-12 and IL-12 + ISG15) or $10^7$ cells/ml (PMA + Ionomycin). Analysis as in (d), n = 4-6, pooled from two independent experiments. g. Blocking of IL18-Ra significantly reduces ISG15 effect. Splenocytes were incubated with aIL-18Ra antibody (10 ug/ml) for 30 minutes and consecutively stimulated with IL-12 and ISG15 for 6h as in Fig 2c and 2d. n = 5, representative of two independent experiments. h. ISG15 stimulates IFN-γ independent of endogenous ISG15. ISG15^−/− and wild-type spleno-cytes were stimulated with IL-12 (20 ng/ml) or a combination of IL-12 and ISG15 (250 ng/mL) for 36h, IFN-γ levels were calculated. Data were analyzed using two-way ANOVA with Šídák's test, n = 4. i. IL-10 levels released by mouse splenocytes induced by ISG15. ISG15^−/− and wild-type splenocytes were stimulated as in (a) for IL-10 production. Analysis as in (a), n = 4. (Statistical significance is indicated as follows: *p ≤ 0.05; **p ≤ 0.01; ***p ≤ 0.001; ****p ≤ 0.0001; ns, not statistically significant).

purified NK cells compared to those in splenocyte suspensions, suggesting that other cytokines or immune cells may contribute to NK cells IFN-γ secretion under these conditions. Since IL-18 has a strong synergistic effect with IL-12, potentiating its ability to induce IFN-γ production by NK and CD8+T cells [49], we evaluated IFN-γ production at different concentrations of IL-18 in the absence or presence of ISG15. Particularly at lower IL-18 concentrations, ISG15 signifi-cantly enhanced the IFN-γ response by NK cells and CD8+T cells in whole splenocyte suspensions (Figs 2c,2d, S3a and S3b), while having minimal effects on IFN-γ production in IL-18-stimulated isolated NK cells (Fig 2e). This led us to hypothesize that ISG15 may influence other cells, facilitating increased IFN-γ production by NK cells through an indirect, cell-extrinsic mechanism. Because of the strong stimulatory effect of IL-18 on NK cells, we tested IL18Ra^fl/fl NKp46^cre ("IL18Ra KO") mice, which specifically lack the IL18 receptor α subunit on NK cells. IL18Ra KO NK cells did not show any increase in IFN-γ production in response to ISG15 addition (Fig 2f). To rule out a general functional defect in the IL18Ra KO NK cells, as has been reported [50], we stimulated the cells with PMA + Ionomycin. The knockout cells exhibited largely intact IFN-γ production, though with a non-significant trend toward a decrease (Fig 2f) [50]. Additionally, using an IL-18Ra-blocking antibody during stimulation with IL-12 and/or ISG15, we confirmed that in the absence of IL-18 signaling, the effect of ISG15 was significantly reduced (Fig 2g). These findings suggest that part of the ISG15 effect on NK cells may be mediated by ISG15-induced IL-18 production or, alternatively, by sensitization to low-level endogenous IL-18 pres-ent in the splenocyte suspensions.

To investigate if endogenous ISG15 is required for the induction of IFN-γ, we stimulated splenocytes from wild-type (WT) and ISG15^−/− mice with ISG15 and found a similar IFN-γ production (Fig 2h), demonstrating that intracellular ISG15 of immune cells is dispensable for IL-12-induced IFN-γ production. Interestingly, we also detected a strong induction of IL-10 in murine splenocytes by extracellular ISG15. ISG15-induced secretion of IL-10 was also independent of intracellu-lar ISG15, as IL-10 secretion was similar in WT and ISG15^−/− splenic immune cells (Fig 2i). Taken together, these *in vitro* assays demonstrate the diverse immunostimulatory roles of extracellular ISG15. Extracellular ISG15 induces outside-in signaling, promoting an IFN-γ response and IL-10 secretion in murine immune cells. Both signaling effects are driven by extracellular ISG15 and are independent of endogenous ISG15.

## Infection-induced secreted ISG15 signals to immune cells

Given the robust induction of ISG15 production and secretion observed in infected human primary cells and in infected mFGT organoids, we next investigated the effect of ISG15 secreted from infected non-immune cells on immune cells. Therefore, we established a co-culture model comprising mFGT organoids in the lower chamber and immune cells in the upper chamber of a transwell plate separated by a 0.4 μm membrane (see schematic diagram in Fig 3a). We estab-lished a co-culture model using WT or ISG15^−/− organoids infected with Ct and enriched NK cells. Recombinant IL-12 was

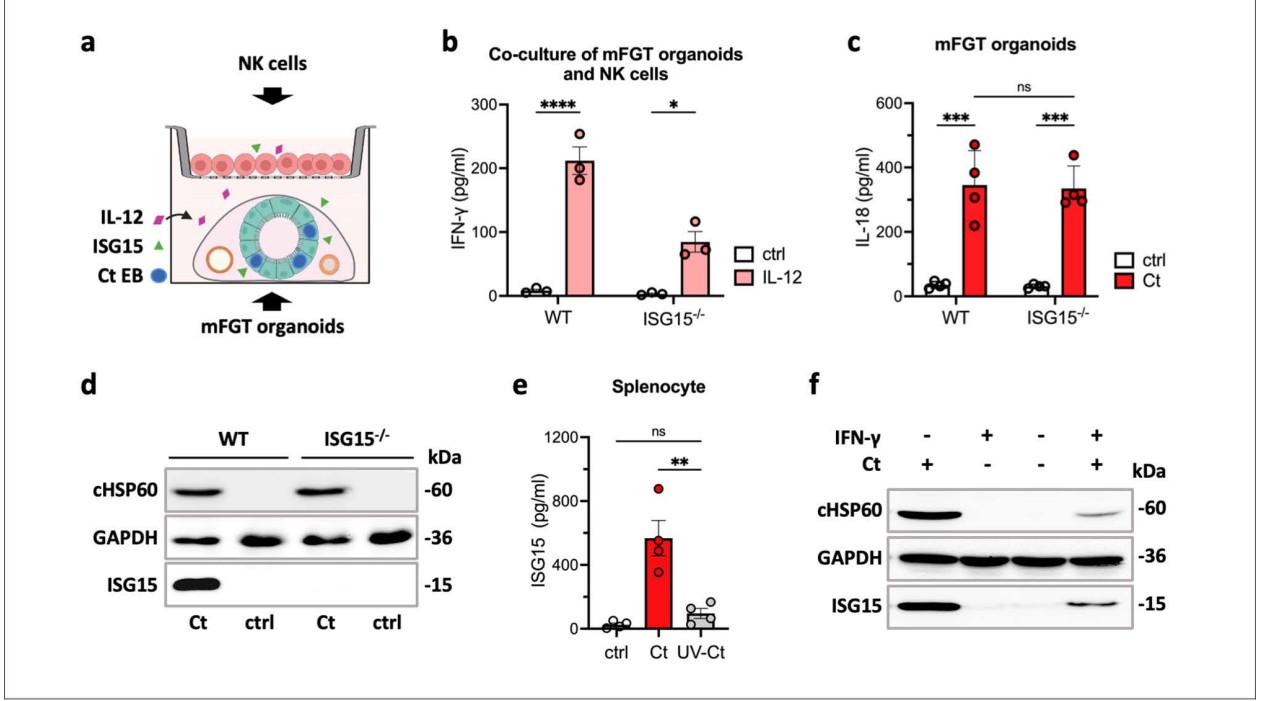

**Fig 3. The effects of secreted ISG15 on immune cells in response to Ct infection.** a. Schematic illustration of the transwell co-culture model with mouse FGT organoids in the lower chamber and mouse NK cells in the upper chamber, separated by a 1.0 µm transwell membrane. b. Infection-induced ISG15 secretion by mFGT organoids increases IFN-γ production by NK cells in the co-culture model. Wild-type and ISG15$^{-/-}$ mFGT organoids infected with Ct were cultured in the lower chambers for 5 days. Freshly isolated mouse NK cells were then seeded on transwell membranes. IL-12 (5 ng/ml) was added simultaneously. IFN-γ levels released by NK cells were measured by ELISA after 36 h of co-culture. Data were analyzed using two-way ANOVA with Tukey's test, n = 3. c. IL-18 secretion in wild-type and ISG15$^{-/-}$ mFGT organoids infected with Ct for 5 days. IL-18 level in supernatant was measured by ELISA. Data were analyzed using two-way ANOVA with Šídák's test, n = 4. d. ISG15 expression in splenocytes infected with Ct. Wild-type and ISG15$^{-/-}$ splenocytes were either left uninfected or were infected with Ct (MOI 2) for 36 h. ISG15, HSP60, and GAPDH expression were analyzed by Western blotting. A representative blot from three independent experiments is shown. e. Viable *Chlamydia* are required to induce efficient ISG15 secretion. Splenocytes were either left uninfected or were infected with Ct (MOI 2) or UV-inactivated Ct (MOI 2) for 36 h. ISG15 secretion was quantified by ELISA. Data were analyzed using one-way ANOVA with Tukey's test, n = 4. f. Persistent *Chlamydia* infection does not induce significant ISG15 expression in host cells. HUVEC cells were either left untreated or were pretreated with 100 ng/mL of IFN-γ for 2 h. Cells were then either left un-infected or were infected with Ct (MOI 1) for 36 h. ISG15, HSP60, and GAPDH expression were analyzed by Western blotting. (Statistical significance is indicated as follows: *p ≤ 0.05; **p ≤ 0.01; ****p ≤ 0.0001; ns, not statistically significant). Fig 3a was created in BioRender (2025; https://BioRender.com/ymuzagl).

added to the corresponding lower chambers which together with the ISG15 secreted by the infected WT organoids should trigger IFN-γ release by NK cells in the upper chamber. Following the addition of IL-12, only modest levels of IFN-γ were detected in the ISG15$^{-/-}$ co-culture system. In contrast, significantly higher IFN-γ levels were observed in the WT organoid co-culture with the addition of IL-12 (Fig 3b). This indicates that the secretion of ISG15 in response to Ct infection by WT organoids enhances the release of IFN-γ by NK cells in the co-culture system. Epithelial cells produce IL-18 in response to Ct infection [51] and we observed elevated IL-18 levels in supernatants of infected mFGT organoids from WT and ISG15$^{-/-}$ mice (Fig 3c). These findings suggest that NK cell activation in this system is driven by both epithelial-derived ISG15 and IL-18.

Furthermore, we investigated the influence of Ct infection on the production of ISG15 in immune cells. ISG15 expression was upregulated in murine splenocytes that had been challenged with Ct (Fig 3d). Interestingly, ISG15 secretion in splenocytes depended on viable *Chlamydia,* since UV-inactivated Ct failed to induce ISG15 secretion in splenocytes (Fig 3e). This finding is consistent with our observations in HUVEC cells, where persistent Ct induced by IFN-γ pretreatment

failed to induce high ISG15 expression (Fig 3f), suggesting that the secretion of ISG15 from infected cells depends on the presence of viable and active Ct. Furthermore, our findings indicate that ISG15 secreted by Ct-infected cells plays a role in signaling that enhances the IFN-γ response of NK cells.

**ISG15 supports protection from *Chlamydia* infection**

To investigate the role of ISG15 in Ct infection *in vivo*, we performed mouse infection experiments in WT and ISG15⁻/⁻ mice. The mice were transcervically infected with EBs for different periods of time (Fig 4a) and the upper genital tract pathology, mainly the occurrence of hydrosalpinx and characteristic uterine horn pathology, was evaluated [52]. At day 14 post infection, the uterine horn pathology in ISG15⁻/⁻ mice was more severe than in WT mice (Fig 4b), although neither hydrosalpinx nor oviduct dilation was observed. We also compared the bacterial loads by culturing supernatants of homogenized FGT tissues and quantifying IFUs of EBs. There was no significant difference in bacterial loads between WT mice and ISG15⁻/⁻ mice at early time points post infection (3 dpi). However, by 7 dpi, ISG15⁻/⁻ mice already showed a trend toward increased bacterial burden compared to WT mice (p = 0.0734), although the difference did not reach statistical significance (Fig 4c). This trend became significant at later time points, with ISG15/⁻ mice exhibiting significantly higher bacterial loads than WT mice at 14 dpi and 28 dpi (Fig 4c). These differences in chlamydial load indicate that ISG15 WT mice are more effective at eradicating the infection and that ISG15 plays an important role in the defense against Ct infection.

The clearance of chlamydial infection is strongly associated with IFN-γ immunity, particularly the production of IFN-γ by innate immune cells. This represents a critical step in immunologically limiting the spread of *Chlamydia*. Therefore, levels of IFN-γ in the FGT tissue of WT and ISG15⁻/⁻ mice were monitored. The basal levels of IFN-γ were found to be comparable between WT and ISG15⁻/⁻ mice in the absence of infection (Fig 4d). Following Ct infection, ISG15⁻/⁻ mice exhibited reduced IFN-γ levels at 3 and 7 dpi, consistent with ISG15 regulating the early, innate IFN-γ response mediated by NK cells (Fig 4d). Consistently, ISG15 levels were strongly increased at this timepoint in infected as compared to uninfected tissue (Fig 4e). Ct infection markedly elevated ISG15 levels in genital tract tissues, especially during the early stages of infection. In contrast to viral [53,54] and parasitic infections [39], which elicit a substantial systemic ISG15 responses characterized by elevated serum levels, ISG15 was undetectable in the serum of mice throughout the whole Ct infection process (Fig 4e). Instead, high levels of ISG15 were detected in FGT lavages from Ct infected mice compared to mock-infected mice (Fig 4f). These results demonstrate that ISG15 is secreted in response to Ct genital tract infection, and functions locally in the infected tissue.

To investigate the distribution of immune cells in the FGT during the early stages of infection, we initially performed fluorescence immunohistochemical staining on mock- and Ct-infected FGT tissue, with a particular emphasis on NK cells as a potential source of IFN-γ in early chlamydial infection [28]. Neutrophils reached the luminal epithelium as early as 12 hpi, and increased numbers of NK cells were observed in the submucosa in close proximity to the epithelial layer (S4 Fig). Therefore, ISG15 secreted by Ct-infected epithelial cells may locally regulate the IFN-γ response of NK cells in direct proximity to the infected epithelial lining. We further performed FACS analysis and found a marked increase in NK cell infiltration (live CD45+, CD3-, CD11b+, NK1.1+, CD49b+) at 24 hpi, with no significant difference between wild-type and ISG15/⁻ (Fig 4g and 4h). These results suggest that ISG15 does not influence NK cell recruitment, but rather modulates their functional response, particularly IFN-γ production, at the site of infection.

The data thus far indicate that ISG15⁻/⁻ mice exhibit reduced efficiency in clearing chlamydial infection. This may be attributed to the established role of ISG15 in amplifying IFN-γ signaling. These mice also showed increased signs of severe inflammatory pathology compared to WT mice. The increased pathology may be attributed to higher chlamydial burdens in ISG15⁻/⁻ mice, or alternatively, may be a consequence of immunopathology resulting from inadequate inflammatory response regulation. Since our previous data demonstrated that ISG15 can induce IL-10 release (Fig 2i), we monitored IL-10 production in the FGT tissues of mice following Ct infection. In contrast to the IFN-γ response (Fig 4d),

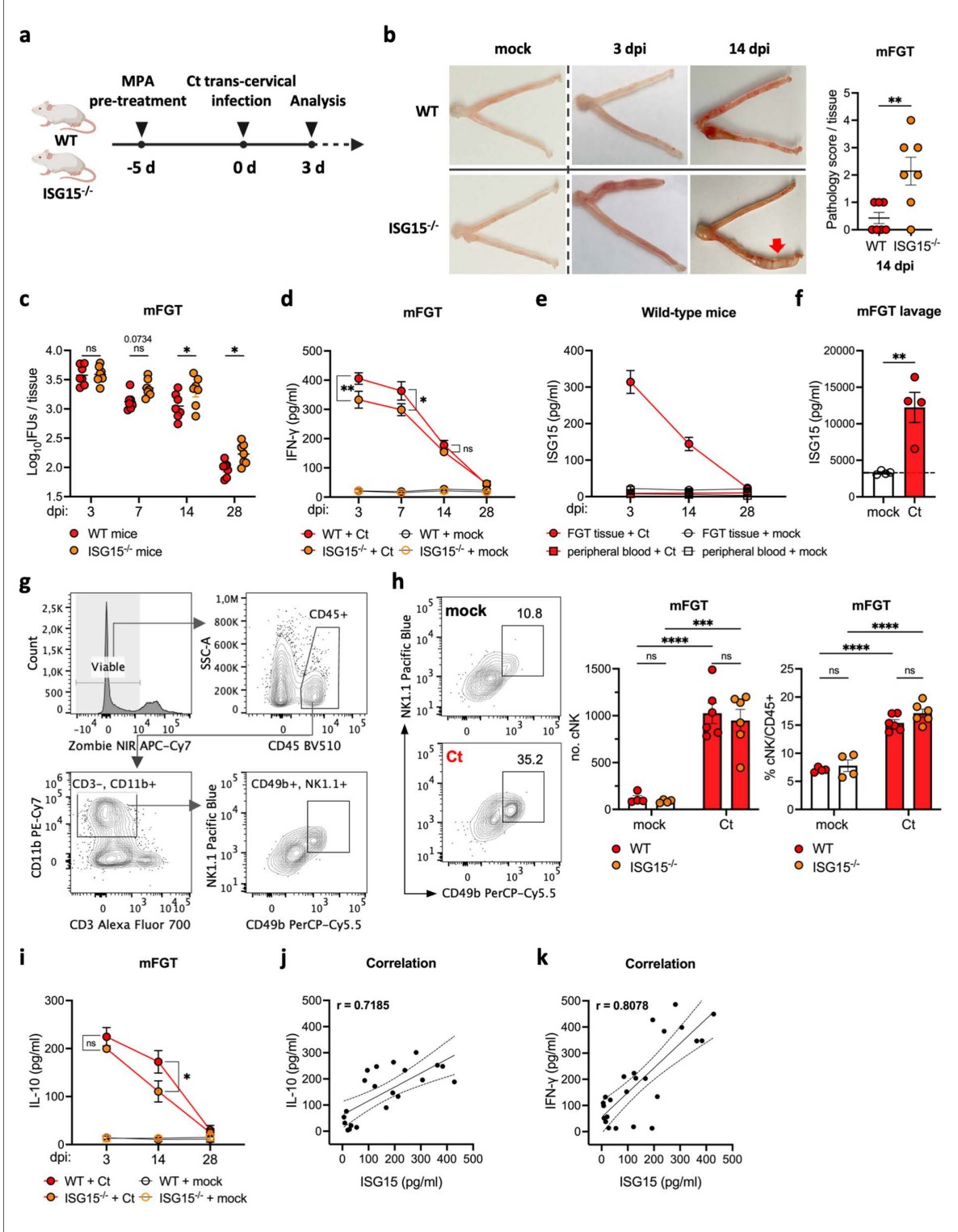

**Fig 4. Secreted ISG15 enhances IFN-γ immunity against Ct in a mouse infection model.** a. Schematic of the Ct infection model. Female C57BL/6J wild-type and C57BL/6J ISG15⁻/⁻ mice were pretreated with medroxyprogesterone acetate (MPA) on day -5 and then infected with Ct EBs by transcervical inoculation on day 0. b. Gross pathology of the upper genital tract in infected wild-type and ISG15⁻/⁻ mice. (Left) Representative images show uterine

horn inflammation at 14 dpi (red arrow). (Right) Quantification of uterine horn pathology scores on 14 dpi; Data were analyzed by Mann-Whitney test, n = 7 per group. c. *Chlamydia* loads in FGT tissues of wild-type and ISG15⁻/⁻ mice. Supernatants of tissue homogenates were transferred to McCoy cells to quantify inclusions after 28 h. Data were analyzed using two-way ANOVA with Šídák's test (n = 7 per group). d. IFN-γ production in FGT tissues of wild-type and ISG15⁻/⁻ mice. IFN-γ levels in the supernatants of tissue homogenates were analyzed by ELISA. Data were analyzed using two-way ANOVA with Tukey's test (n = 7 per infected group, n = 3 per mock group). e. ISG15 levels in FGT tissues and serum from infected wild-type mice, determined by ELISA (n = 7 per infected group, n = 3 per mock group). f. ISG15 levels in FGT lavages in Ct-infected wild-type mice. ISG15 levels in FGT lavages were measured by ELISA on day 3 pi in Ct-infected and mock-infected wild-type mice. Data were analyzed by unpaired t-test, n = 4. g. Representative FACS plots illustrating the gating strategy used to identify NK cells in FGT tissue. h. NK cell infiltration into the FGT 24 hpi. (Left) Representative FACS plots showing NK cell population from WT mice infected with Ct and mock-infected. (Right) Quantification of NK cell number and frequency (as % of CD45 + cells) in WT and ISG15⁻/⁻ mice. Data were analyzed using two-way ANOVA with Šídák's test. (n = 6 per infected group, n = 4 per mock group). i. IL-10 production in FGT tissues of wild-type and ISG15⁻/⁻ mice. Supernatants of tissue homogenates from d. were analyzed by IL-10 ELISA. Data were analyzed using two-way ANOVA with Tukey's test. j. The correlations between ISG15 levels and IL-10 levels in the FGT tissues of infected wild-type mice. The value r represents Pearson's correlation coefficient, n = 21. k. The correlations between ISG15 levels and IFN-γ levels in the FGT tissues of infected wild-type mice, n = 24. (Statistical significance is indicated as follows: *p ≤ 0.05; **p ≤ 0.01; ns, not statistically significant; n indicates independent biological replicates). Fig 4a was created in BioRender (2025; https://BioRender.com/14hsbor).

IL-10 levels were comparable in WT and ISG15⁻/⁻ mice at early infection time points (3 dpi). However, at 14 dpi, IL-10 levels were statistically lower in ISG15⁻/⁻ mice compared to WT mice, despite the increased bacterial burden (Fig 4c and 4i). The modulation of both, IFN-γ and IL-10 secretion by ISG15 was further confirmed by correlating their levels in the FGT tissues of infected wild-type mice. We found a significant positive correlation between ISG15 and IFN-γ, with a Pearson correlation coefficient of 0.8078, and between ISG15 and IL-10, with a coefficient of 0.7185 (Fig 4j and 4k). These analyses support the assumption that the immunomodulatory effects of ISG15 that we observed *in vitro*, were also active during infection *in vivo*.

## ISG15 restrains IFN-γ-induced proinflammatory macrophage polarization

During the acute phase of infection, excessive inflammatory responses need to be limited to prevent severe tissue damage [55]. Macrophages adopt distinct phenotypes in response to the inflammatory microenvironment, with IFN-γ and IL-10 being key polarizing signals [56]. IFN-γ drives pro-inflammatory macrophage polarization to combat infection [57,58]; however, aberrant IFN-γ stimulation is associated with immunopathology in inflammatory diseases, such as pneumococcal meningitis and macrophage activation syndrome [59,60]. In contrast, IL-10 initiates suppressive mechanisms that downregulate the inflammatory response and reduce pathological tissue damage during infection [55,61–63]. ISG15⁻/⁻ mice exhibited exacerbated inflammatory pathology and low levels of IL-10 in the FGT at the later stage of infection (Fig 4), suggesting that ISG15-induced IL-10 limits pathology during infection. To investigate whether ISG15 also acts directly on macrophages, we analyzed the polarization of murine bone marrow-derived macrophages (BMDMs, Mφ) after *in vitro* stimulation with ISG15. This included the M1 subtype stimulated with IFN-γ and the M2 subtypes stimulated with IL-4 or IL-10 as classical controls, in comparison to untreated BMDMs as M0 Mφ (Fig 5a). Given the heterogeneity of M2 macrophages [64], macrophage polarization requires the integration of multiple surface markers and functional outputs. Therefore, we analyzed the expression of surface markers, specifically CD86, CD163, and CD206 in the gated macrophage population (live CD11b+ , F4/80+) (S5a Fig). As expected, IFN-γ induced a pro-inflammatory phenotype in macrophages, characterized by high frequency of CD86+ cells and CD86 expression (Fig 5b and 5c). Conversely, stimulation with IL-10 and IL-4 resulted in a high frequency of CD206+ cells and CD206 expression (Fig 5b and 5d). In contrast, macrophages stimulated with ISG15 showed significantly higher frequency of CD163+ cells and CD163 expression (Fig 5b and 5e), which is characteristic of M2-like macrophages. Furthermore, ISG15⁻/⁻ macrophages showed similar polarization responses (S5b and S5c Fig), and ISG15 stimulation of ISG15⁻/⁻ macrophages also elevated the CD163+ population (Fig 5f), indicating that ISG15 can directly reprogram macrophage polarization into an M2-like population through an outside-in signaling mechanism, independent of intracellular ISG15 (Fig 5g).

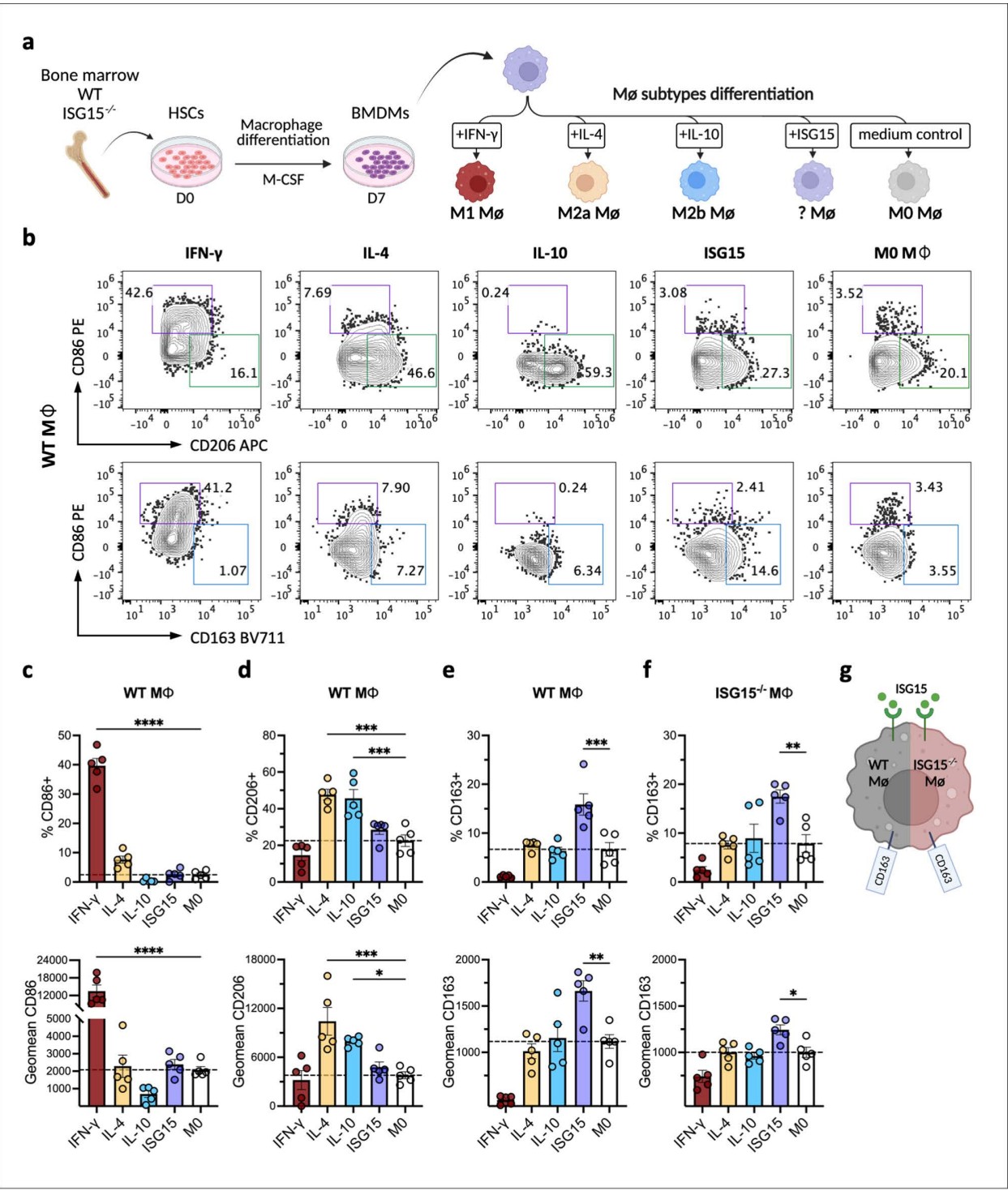

**Fig 5. ISG15 induces polarization of wild-type and ISG15$^{-/-}$ macrophages.** a. Schematic illustration of the differentiation and subtypes of BMDMs. Bone marrow cells were isolated from wildtype and ISG15$^{-/-}$ mice and cultured with M-CSF stimulation for 7 days, during which hematopoietic stem cells (HSCs) developed into macrophages (BMDMs). BMDMs were further treated with cytokines to analyze subtypes. b. Representative FACS plots characterizing macrophage polarization. Wild-type BMDMs (WT Mφ) were stimulated with IFN-γ, IL-4, IL-10, or ISG15 (40 ng/ml) for 24 h, along with M0 Mφ without stimulation. The expression of surface markers CD86, CD206, and CD163 was analyzed by FACS in the macrophage population (live CD11b+, F4/80+ cells). c. The frequency of CD86+ cells and the Mean Fluorescence Intensity (MFI) of CD86 on gated wild-type Mφ (live CD11b+,

F4/80+) stimulated as described in (b). MFI is presented as Geomean values, data were analyzed using one-way ANOVA, n = 5. d. The frequency of CD206+ cells and the MFI of CD206 on gated wild-type Mɸ stimulated as described in (b). Data were analyzed using one-way ANOVA, n = 5. e. The frequency of CD163+ cells and the MFI of CD163 on gated wild-type Mɸ stimulated as described in (b). Data were analyzed using one-way ANOVA, n = 5. f. The frequency of CD163+ cells and the MFI of CD163 on gated ISG15$^{-/-}$ BMDMs (ISG15$^{-/-}$ Mɸ) stimulated as described in (b). Data were analyzed using one-way ANOVA, n = 5. g. Schematic illustrating that extracellular ISG15 regulates macrophage polarization through an outside-in signaling mechanism, independent of intracellular ISG15. (Statistical significance is indicated as follows: *p ≤ 0.05, **p ≤ 0.01; ***p ≤ 0.001; ****p ≤ 0.0001; ns, not statistically significant; n indicates independent biological replicates. Fig 5a and 5g were created in BioRender (2025; https://BioRender.com/ag9c78z; https://BioRender.com/ciedv2b).

As *Chlamydia* infection is kept in check by IFN-γ, and prolonged IFN-γ production is a hallmark also of chronic *Toxoplasma gondii* infection [65], IFN-γ is likely to play a role in the macrophage polarization during Ct infection. We therefore tested whether ISG15 could act to limit the pro-inflammatory macrophage polarization induced by IFN-γ. To this end, we examined the polarization characteristics of macrophages during Ct infection under stimulation with IFN-γ or IFN-γ/ISG15. In the presence of IFN-γ, infected macrophages exhibited a polarized state, characterized by a population with high CD86 expression (CD86$^{hi}$). However, the addition of ISG15 reduced this CD86$^{hi}$ population while increasing the CD163+ population (Fig 6a). The statistical analysis demonstrated that the frequency of the CD86$^{hi}$ population was significantly reduced in the IFN-γ/ISG15 group in comparison to the IFN-γ group, while the frequency of the CD163+ population was significantly increased (Fig 6b and 6c). The population with intermediate CD86 expression (CD86$^{int}$) remained prominent in the IFN-γ/ISG15 group, despite a significant reduction in its frequency (Fig 6d). MFI analysis further confirmed a significant decrease in overall CD86 expression and a significant increase in CD163 expression in the ISG15/IFN-γ group (Fig 6e). To further functionally evaluate these polarized phenotypes of macrophages adapted to IFN-γ or IFN-γ/ISG15 conditions, we measured cytokine production levels. Importantly, IL-10 secretion was significantly induced in the ISG15/IFN-γ treated group (Fig 6f), whereas the levels of the pro-inflammatory cytokines IL-6 and TNF-α, which drive acute inflammatory response, were significantly reduced (Fig 6g and 6h). These data provide compelling evidence that ISG15 plays a pivotal role in macrophage polarization, influencing the conversion of macrophages that are defending against Ct exposed to IFN-γ into a less aggressive state with increased IL-10 and decreased inflammatory cytokine production.

The high levels of ISG15 induced locally in the FGT in response to Ct infection in WT mice, the exacerbated pathology and lower IL-10 levels observed in the FGT of infected ISG15$^{-/-}$ mice, and the evidence that ISG15 not only promotes an M2-like phenotype in macrophages but also counteracts the enhanced inflammatory response of polarized M1 macrophages *in vitro*, support the critical role of ISG15 in regulating inflammatory responses during chlamydial infection. These findings demonstrate that ISG15 plays a pivotal role in facilitating the clearance of Ct and preventing macrophages from triggering aggressive inflammatory reactions. This is achieved by mediating their polarization and cytokine responses in genital tract infections.

## Discussion

Ct infections of the female genital tract can lead to severe pathologic outcomes, predominantly due to the inflammatory response. The secretion of IFN-γ in response to infection is a central event in the immunological defense against local *Chlamydia* infection and in limiting the spread of infection. However, there is limited information on the mechanisms underlying the IFN-γ response in the FGT at the one site and the orchestration of subsequent inflammatory progression at the other site. Our data show that ISG15 plays a central role at the interface of IFN-γ-induced immune defense and regulation of inflammation.

ISG15 contributes to host resistance to viral and bacterial infections, either by covalent modification of target proteins, or as a secreted cytokine associated with IFN-γ responses [40,43,44]. Infection of human primary epithelial and endothelial cells and mouse FGT organoids with Ct and Cm induced high protein expression and secretion of ISG15 (Figs 1 and S1). This is in line with previous transcriptional analyses in epithelial cells [45,46]. An interesting observation was that type

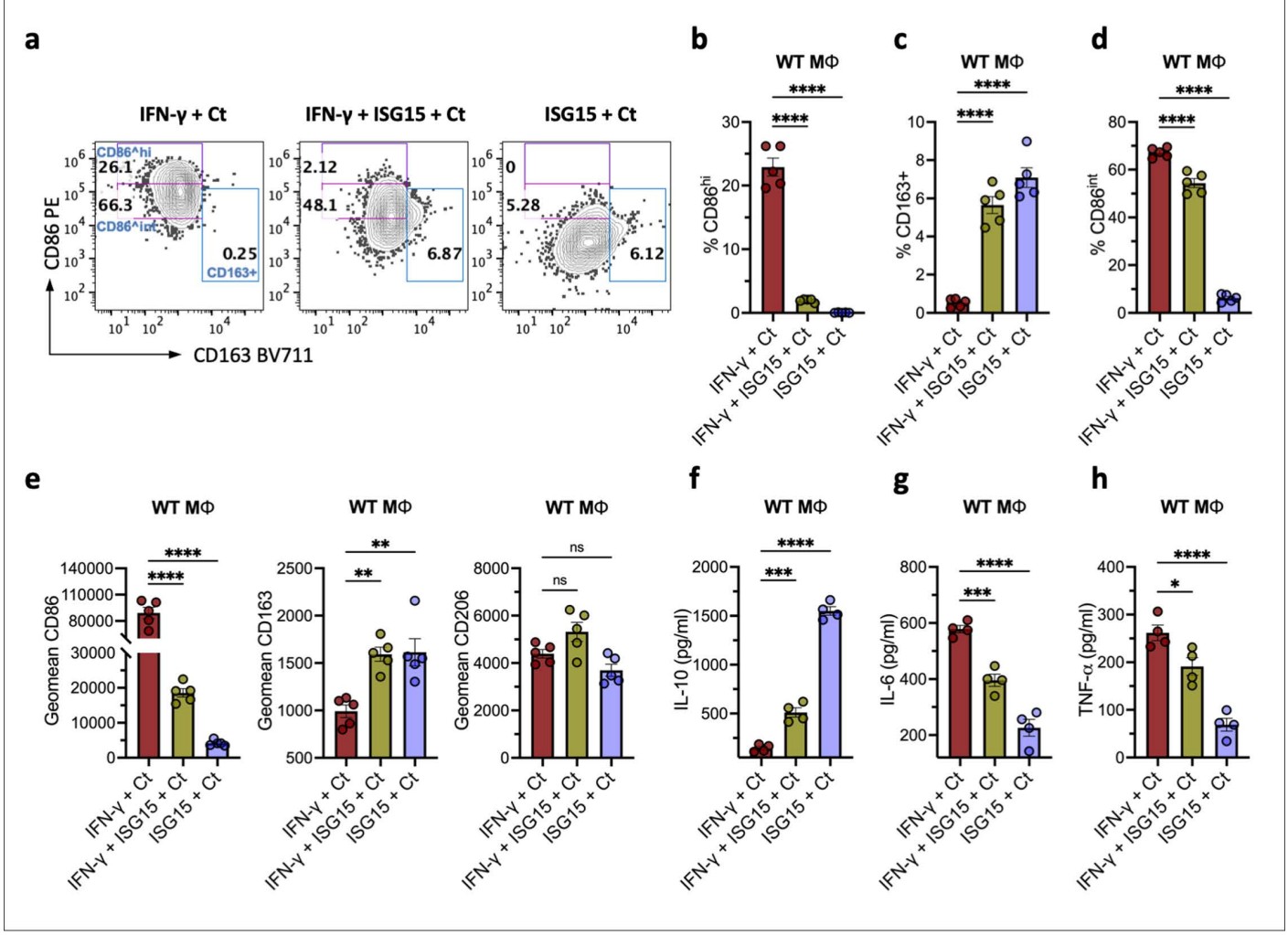

**Fig 6. ISG15 downregulates IFN-γ-mediated over-polarization of M1 macrophages and the inflammatory response.** a. Representative FACS plots characterizing polarized macrophages upon Ct infection. Wild-type macrophages (WT Mφ) were pretreated with either IFN-γ (40 ng/ml), ISG15 (40 ng/ml), or IFN-γ + ISG15 for 2 h, then infected with Ct (MOI 5) for 24 h. The CD86hi, CD86int and CD163+ cells in the WT Mφ population (live CD11b+, F4/80+) were analyzed by FACS. b. The frequency of CD86hi cells in gated WT Mφ (live CD11b+, F4/80+) stimulated as described in (a). Data were analyzed using one-way ANOVA, n = 5. c. The frequency of CD163+ cells in gated WT Mφ stimulated as described in (a). Data were analyzed using one-way ANOVA, n = 5. d. The frequency of CD86int cells in gated WT Mφ stimulated as described in (a). Data were analyzed using one-way ANOVA, n = 5. e. MFI of CD86, CD163, or CD206 in gated WT Mφ (live CD11b+, F4/80+) treated as described in (a). Data were analyzed using one-way ANOVA, n = 5. f. IL-10 levels secreted by Mφ infected with Ct under IFN-γ+/- ISG15 stimulation. WT Mφ were pretreated with either IFN-γ (40 ng/ml), ISG15 (40 ng/ml), or IFN-γ + ISG15 for 2 h, then infected with Ct (MOI 5) for 32 h. Cell-free supernatants were analyzed for IL-10 production. Data were analyzed using one-way ANOVA, n = 4. g. IL-6 levels secreted by Mφ infected with Ct under IFN-γ+/- ISG15 stimulation. Assay and analysis were performed as in (f), n = 4. h. TNF-α levels secreted by Mφ infected with Ct under IFN-γ+/- ISG15 stimulation. Assay and analysis were performed as in (f), n = 4. (Statistical significance is indicated as follows: *$p \leq 0.05$; **$p \leq 0.01$; ***$p \leq 0.001$; ****$p \leq 0.0001$; ns, not statistically significant; n indicates independent biological replicates).

I IFN induced the expression of ISG15 protein in all cells tested, but significant secretion of ISG15 was only observed in response to infection and not to type I IFN stimulation. In contrast to our findings of a strong Ct-induced expression and secretion of ISG15 only in primary cells, a recent study demonstrated expression but no secretion of ISG15 in infected HeLa cells [66]. The authors therefore concluded that ISG15 exerted its control from an intracellular location, rather than

as a secreted cytokine. Although we cannot exclude a role of intracellular ISG15 during *Chlamydia* infection, the effects described here on the dual role of ISG15 in pro- and anti-inflammatory signaling depends on secreted ISG15. This is in line with elevated ISG15 levels in FGT tissues and in FGT lavages from infected mice. Interestingly, in contrast to viral and parasitic infections, ISG15 levels were not elevated in the serum of infected mice (Fig 4e and 4f). This suggests that ISG15 exerts its immunoregulatory function locally in the genital tract. This is supported by the observed defect of ISG15$^{-/-}$ mice to clear infection as efficiently as WT mice (Fig 4c). Differences in chlamydial burden between knockout and WT mice were significant at later rather than at earlier stages of infection. This is consistent with a role for ISG15 in the clearance of infection rather than the susceptibility of the host to infection.

Consistent with the effects of extracellular ISG15 on human immune cells reported by others [41,48], our stimulation assay demonstrated a strong synergistic impact of ISG15 and IL-12 on IFN-γ secretion by NK and T cells. (Figs 2a and S3). However, this effect was most pronounced when NK cells were stimulated in a whole splenocyte suspension compared to isolated NK cell cultures, suggesting a role for bystander cells (Fig 2b). This was additionally dependent on the ability of NK cells to sense IL-18, suggesting a partially indirect effect of ISG15 inducing secretion of IL-18 or sensitizing NK cells to low amounts of endogenous IL-18 (Fig 2c and 2d). Using co-culture models of mouse FGT organoids and NK cells, we provided direct evidence that ISG15 secreted by infected genital epithelial cells enhances IFN-γ production by NK cells (Fig 3a–3c). Secreted ISG15 exerts a similar synergistic effect together with IL-18, enhancing the IFN-γ response in NK cells particularly at lower concentrations of IL-18 (Fig 2c). This suggests that ISG15 supports IFN-γ production of NK cells in an environment with limited access to IL-18. We believe that this synergistic role of ISG15 is important in the early stages of host infection, particularly in situations where networks of different immune cells are not yet well-established in the infected tissue, or immune cells are not fully activated. During this phase of infection, when levels of pro-inflammatory cytokines (such as IL-18) are low, secreted ISG15 from infected epithelial cells can help NK cells to increase secreted IFN-γ levels, thereby enabling an earlier and more rapid response to the infection.

The rapid accumulation of NK cells in the subepithelial layer and around the epithelial lining of the infected FGT was detected as early as 12 hpi (S4 Fig). This feature of NK cell localization further supports our hypothesis that secreted ISG15, induced by Ct infection in epithelial cells, signals to nearby NK cells to enhance the IFN-γ response. The observation of reduced IFN-γ levels in FGT tissues from ISG15$^{-/-}$ mice compared to wild-type mice, evident as early as 3 dpi (Fig 4d), supports our hypothesis. Interestingly, the induction of IFN-γ secretion by immune cells was dependent on secreted ISG15, as ISG15$^{-/-}$ immune cells also efficiently produced IFN-γ when stimulated with ISG15 (Fig 2h). We therefore hypothesize that ISG15$^{-/-}$ mice clear Ct less efficiently due to the absence of the potentiating effect of secreted ISG15 on the IFN-γ production, and therefore have a higher bacterial burden. Specifically, ISG15 exerts cell-to-cell signaling locally in the genital tract, from infected epithelial cells to nearby NK cells, and synergizes with IL-12 and IL-18 to enhance IFN-γ immunity against Ct. Although IFN-γ is not the sole contributor to bacterial clearance, our findings support a critical role for ISG15 in initiating early IFN-γ responses that shape the course of infection. Notably, our data showed that persistent Ct infections induced by IFN-γ treatment did not result in high levels of ISG15 (Fig 3f), indicating that the ISG15-mediated innate immune response is primarily directed against active Ct infections, rather than persistent infections.

It is a normal physiological process for the host to mount an inflammatory response to pathogens during infection. However, uncontrolled inflammatory responses lead to increased inflammation and consequently to tissue damage, which is one of the hallmarks of immunopathological processes. In this study, we observed that mice lacking ISG15 exhibited uterine horn pathology and lower levels of IL-10 in FGT tissue at a later stage of infection (Fig 4b and 4i). Macrophages play a key role in both pro- and anti-inflammatory responses, and their phenotypes are highly responsive to the inflammatory microenvironment. M1-like macrophages activated by LPS and IFN-γ, have pro-inflammatory and bactericidal functions. In contrast, M2-like macrophages have anti-inflammatory functions that promote tissue repair and restoration of tissue homeostasis [17,56]. During infection, as the pathogen load decreases, M1-like macrophages undergo a switch to an M2-like phenotype, which inhibit excessive immune activation. This is achieved primarily by the production of high levels of anti-inflammatory

cytokines such as IL-10, which ultimately restores a homeostatic balance between M1 and M2 macrophages [67–70]. Our study shows that ISG15 downregulates and controls Ct-induced inflammation in genital tract infections. This could be achieved by directly programming macrophages into a M2-like CD163+ phenotype and promoting IL-10 release (Figs 5,6a and 6f). This regulatory effect of ISG15 turns the high inflammatory response of macrophages, which have the potential to cause tissue damage, into a low inflammatory state (Fig 6). While our *in vitro* findings provide strong support for a modulatory role of ISG15 in macrophages, further *in vivo* investigation is needed to clarify the relevance of this effect during infection. A previous study [48] described an anti-inflammatory ISG15/IL-10 axis in other disease contexts, supporting our findings. However, the interplay between ISG15 and IL-10 within the *in vivo* cytokine network remains to be fully elucidated. Given the plasticity of macrophages, they exhibit opposing functions that either stimulate or down-regulate inflammation. We hypothesize that ISG15 secreted from the site of infection is important for reducing excessive deleterious inflammation in chlamydial infection. The enhanced pathological effects observed in ISG15$^{-/-}$ mice may be a combined result of the higher Ct load and the lack of ISG15 to balance the macrophage polarization.

In conclusion, our data demonstrate that FGT epithelial cells locally secrete ISG15 in response to Ct infection and that this secreted ISG15 plays distinct roles during chlamydial infection of the genital tract. ISG15 acts synergistically with IL-18 cytokine to increase the IFN-γ production by NK cells early after infection, which is critical for the rapid establishment of innate IFN-γ immunity and efficient clearance of the bacterial pathogen. Subsequently, ISG15 exerts an immunomodulatory role by regulating macrophage polarization, reducing macrophage responsiveness to IFN-γ as an M1-polarizing signal, thereby effectively controlling the progression of inflammation in chlamydial genital tract infection (Fig 7).

## Methods

### *Chlamydia* strain and purification of infectious EBs

In this study, we used the *Chlamydia trachomatis* serovar L2/434/Bu (Ct) strain (ATCC VR-902B) and *Chlamydia muridarum* Nigg II (Cm) strain (ATCC VR-123). Ct were grown in HeLa 229 cells at a multiplicity of infection

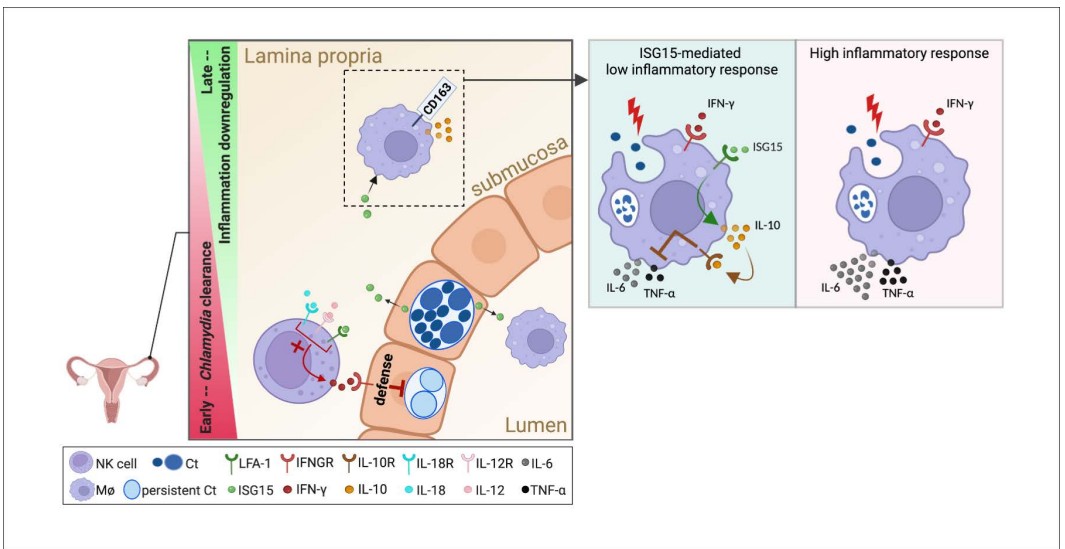

**Fig 7. Secreted ISG15 modulates the immune response during Ct FGT infection.** Schematic representation of the immunomodulatory functions of ISG15 in the FGT during Ct infection. After Ct infection, genital epithelial cells release ISG15. ISG15, in synergy with IL-12 and IL-18, potentiates the production of IFN-γ by NK cells, contributing to the immune defense against the infection. In addition, the ISG15 exerts regulatory effects on macrophages to downregulate the inflammatory response. Created in BioRender. (2025; https://BioRender.com/a9cz4u7).

(MOI) of 1. The bacteria were harvested in cold 1x sucrose-phosphate-glutamic acid (SPG) buffer (7.5% sucrose, 0.052% $KH_2PO_4$, 0.122% $Na_2HPO_4$, 0.072% L-glutamate, pH 7.2) by mechanically lysing the infected HeLa 229 cells at 48 hours post-infection (hpi) using glass beads. Cell lysates were centrifuged at 2,000 x g for 10 minutes at 4°C to separate Ct from cell debris. Ct were then harvested via centrifugation at 35,000 x g for 30 minutes at 4°C. Ct EBs were purified via 3-layer renografin (Bayer, PZN# 00408497) gradients (13 ml 20%, 8 ml 30%, 5 ml 40%) by centrifugation at 18,000 rpm, 4°C, for 90 minutes. The pellets (Ct EBs) were resuspended in 1 x SPG buffer, aliquoted, and stored at -80°C until further use. To titrate the EBs stock, thaw one vial and make 2-fold serial dilutions of the stock. Inoculate the diluted inoculum onto 24-hour-old HeLa 229 cells in 24-well plates. The 24-well plates with inoculum were cultured for 24 hours, after which inclusions were counted at different dilutions and calculated as inclusion-forming units (IFUs) per ml of the original EBs stock. Cm EBs were prepared using the same procedure.

## Cell culture

HeLa 229 cells (ATCC CCL-2.1), McCoy cells (ATCC Cat# CRL-1696, RRID:CVCL3742), and Fimb cells (Human ovarian surface epithelial fimbriae cell, isolated from the fimbriae of patients undergoing hysterectomy) were cultured in RPMI-1640 medium (Gibco) supplemented with 10% (v/v) heat-inactivated FBS (Sigma-Aldrich). HUVEC cells (Human umbilical vein endothelial cells, ATCC CRL-1730) were cultured in Medium 200 (Gibco) supplemented with 10% LSGS (Low Serum Growth Supplement, Gibco). All cells were grown in a humidified incubator with 37°C, 5% v/v $CO_2$. All cells after infection with Ct were maintained in a humidified incubator with 5% (v/v) $CO_2$ at 35°C.

Primary ectocervical epithelial cells (EcCxEp cells) were isolated from biopsies obtained from patients undergoing hysterectomy at the University Hospital, Würzburg, Germany. The biopsies were processed within 24 hours after sampling. EcCxEp cells were cultured in Advanced DMEM/F12 (ThermoFisher, Cat# 12634028) supplemented with 12mM HEPES (ThermoFisher, Cat# 15630080), 1% GlutaMAX (ThermoFisher, Cat# 35050061), 1% B27 (ThermoFisher, Cat# 17504-001), 1% N2 (ThermoFisher, Cat# 17502001), 10ngml$^{-1}$ human EGF (Peprotech, Cat# AF-100-15), 100ngml$^{-1}$ human noggin (Peprotech, Cat# 120-10C), 100ngml$^{-1}$ human FGF10 (Peprotech, Cat#100-26-500), 1.25mM N-acetyl-L-cysteine (Sigma-Aldrich, Cat# A9165-5G), 10mM nicotinamide (Sigma-Aldrich, Cat# N0636), 2μM TGF-β receptor kinase inhibitor IV (Bio-techne, Cat# 2939/50), 10μM Rock Inhibitor (Y-27632) (Hoelzel, Cat# 1293823), 10μM Forskolin (Bio-techne, Cat# 1099) and 1% penicillin-streptomycin (Sigma Aldrich, Cat# P4333).

## Reagents

In this study recombinant proteins used for stimulation assays include mouse ISG15 (Abbexa, Cat# abx167925), mouse IL-12 (R&D, Cat# 419-ML-010/CF), mouse IL-10 (Peprotech, Cat# 210-10-2UG), mouse IL-4 (Biolegend, Cat# 574302), mouse IL-18 (Biolegend, Cat# 767002), mouse ISG15 (LifeSpan Biosciences, Cat# LS-G26127), mouse IFN-gamma (IFN-γ) (Biolegend, Cat# 575304), human IFN-alpha (IFN-α) (Sigma Aldrich, Cat# IF007), human IFN-gamma (Sigma Aldrich, Cat# 11040596001).

## Isolation of mouse splenocytes

8-10 weeks old mice were euthanized, and spleens were aseptically harvested. The spleens were rinsed three times with 1X HBSS (Hank's Balanced Salt Solution, no calcium, no magnesium, Gibco), and then carefully minced into small pieces with disinfected scissors. Spleen pieces were transferred into a 70 μm cell strainer (Corning) fitted on a 50 ml Falcon tube, gently pressed spleen tissue pieces through the strainer using a flat-end syringe plunger, while continuously adding ice-cold RPMI/10% FBS medium (R10 medium). Cells were collected at 600 rpm, 10 min at 4°C and then resuspended in 2 mL ACK lysis buffer (Gibco) to eliminate red blood cells (RBCs). The cell pellet was resuspended gently in 1 ml of R10 medium after centrifugation. The cell suspension was filtered through a 70 μm cell strainer to remove dead cell clumps

and debris. Cells were counted on a hemocytometer. Isolated splenocytes were either immediately seeded onto 12 well plates with 1.5 x 10^6 cells/well and rested for 2 h within R10 medium in an incubator until further treatment or infection, or directly used further for purification of primary NK cells.

## Purification and verification of NK cells

The murine primary NK cells were enriched from isolated splenocytes using Mouse NK Cell Isolation Kit (STEMCELL Technologies, Cat# 19855 or Miltenyi Biotec, Cat# 130-115-818) according to the manufacturer's protocol. In brief, the negative selection kit contains a concoction of antibodies against cells other than NK cells, like B cells, T cells, and monocytes, therefore isolating untouched and highly purified NK cells from splenocytes via immunomagnetic negative selection. The viability of enriched NK cells was evaluated by trypan blue exclusion method on a hemocytometer, and the purity was verified via FACS assay staining cells with FITC anti-mouse NK1.1 antibody (Biolegend, Cat# 156507) and APC anti-mouse CD3 antibody (Biolegend, Cat# 100235). Afterwards, NK cells were immediately seeded onto a transwell membrane to set up a transwell co-culture model or used for *in vitro* stimulations.

## *In vitro* NK cell stimulations

Splenocytes or isolated NK cells were resuspended in RPMI medium (Thermo Fisher, Cat# 61870044) + 10% FCS + 1% Pen/Strep (Sigma-Aldrich, Cat# P0781-100ML) + 50 uM β-mercaptoethanol (Thermo Fisher, Cat# 31350010) at a density of $10^6$ cells/ml (splenocytes) or $1–5 \times 10^5$ cells/ml (isolated NK cells). Cells were stimulated with one or several of the following: IL-12 (20 ng/ml), IL-18 in concentrations as indicated, ISG15 (250 ng/ml), PMA (100 ug/ml, Sigma-Aldrich, Cat# P8139-1MG) + Ionomycin (250 ug/ml, Sigma-Aldrich, Cat# I0634-1MG) or IL-10 (20 ng/ml) for 4–6 hours. For IL-18Ra blocking experiments, the cells were incubated with IL-18Ra blocking antibody (R&D, Cat# AF856, 10ug/ml) for 30 minutes before addition of cytokines and/or ISG15. Samples were consecutively stained for FACS analysis using the BD Cytofix/Cytoperm for intracellular staining (BD, Cat# 554714). Fc-Block and viability dyes were used throughout the paper to exclude dead cells and unspecific binding.

For all FACS staining, in general: cells were centrifuged at 350 g for 5 min at 4°C and the supernatant was discarded, cells were resuspended in PBS with fixable viability dye and Fc-Block and incubated 10–20 min at 4°C. Cells were washed as above, stained against surface antigens for 20–40 min at 4°C in FACS Buffer containing 0.5% BSA (Roth, Cat # 8076.3), 2mM EDTA (Roth, Cat # 8043.2) + 0.01% sodium azide (Roth, Cat # K305.2) in PBS. They were subsequently fixed for 20 min at 4°C in the indicated fixation buffer and intracellular staining was performed for 30 min at room temperature or overnight at 4°C. Cells were finally washed again and acquired.

Antibodies used: CD16/32 (BioXcell, Cat# BE0307, 1:200), CD19 biotin (eBioscience, Cat# 13-0913-86, 1:200), CD25 PE-Cy7 (Biolegend, Cat# 102016, 1:100), CD3e biotin (Biolegend, Cat# 103204, 1:200), CD3e BV421 (Biolegend, Cat# 145-2C11, 1:200), CD44 BV605 (Biolegend, Cat# 103047, 1:400), CD45 FITC (Biolegend, Cat# 103108, 1:400), CD5 biotin (Biolegend, Cat# 100604, 1:500), CD8a BV785 (Biolegend, Cat# 100750, 1:400), F4/80 biotin (Biolegend, Cat# 123106, 1:300), FceRIa biotin (Biolegend, Cat# 134304, 1:300), IFN-γ APC (Biolegend, Cat# 505810, 1:400), Fixable Viability Dye eF780 (Invitrogen, 65-0865-14, 1:1000), IL-18Ra (R&D Systems, Cat# AF856, 10 ug/ml), NK1.1 (Biolegend, Cat# 108708, 1:300), Streptavidin V500 (BD, Cat# 561419, 1:1200), TCRb biotin (Biolegend, Cat# 109204, 1:300), TCRgd biotin (eBioscience, Cat# 13-5711-85, 1:300).

## Generation and cultivation of mouse female genital tract 3D organoids

After isolation of the murine female genital tract (mFGT) tissues (without vagina) from the ISG15^{−/−} mice and the wild-type mice, mFGT tissue was rinsed with sterile DPBS (Gibco) 3 times, placed into a sterile Petri dish (Corning), and cut into small pieces (1~2 mm) with a disinfected surgical blade. Tissue pieces were transferred into 1 mg/mL collagenase solution (Sigma Aldrich) at 37°C incubation with mild shaking for 60 min. Cell suspension was then filtered

through a 70 μm cell strainer, cells were collected via centrifugation at 1,000 rpm for 5 min at 4°C. The cell pellet was resuspended in Matrigel (Corning) and plated 50 μL/drop for each well in 24-well plates. Add 500 μl organoid growth medium to the wells. Organoid growth medium was made up of 40% Adv DMEM (Thermo Scientific), 25% conditioned Wnt3A medium, 25% conditioned RSPO1 medium, 10% conditioned Noggin medium, with addition of 2% B27 (Thermo Scientific), 1 mM nicotinamide (Sigma Aldrich), 100 ng/ml human FGF10 (Thermo Scientific), 50 ng/ml mouse EGF (Thermo Scientific), 10 μM ROCK inhibitor (Abmole Bioscience), 0.5 μM TGF-β RI Kinase Inhibitor IV (Calbiochem).

ISG15$^{-/-}$ mFGT organoids and WT mFGT organoids were passaged biweekly at a 1:2 ratio. The organoids were mechanically fragmented using a syringe and a 26G needle, centrifuged at 1,000 g at 4°C for 5 min. The collected cells were resuspended with fresh Matrigel and dropwise plated into 24-well plates. Organoids were kept in a humidified incubator with 5% (v/v) $CO_2$ at 37°C.

## Ct infection of mFGT organoids

mFGT organoids were released from Matrigel using ice-cold DPBS, centrifuged (1,000 g, 4°C for 5 min) and resuspended in ice-cold DPBS for a secondary wash to remove the Matrigel completely. Pelleted organoids were then mechanically fragmented by passing through a 26G needle. Fragmented organoids were distributed in equal amounts into 1.5 ml Eppendorf tubes and were then infected with Ct EBs (1 × 10$^6$ IFUs per sample) for 60 min at 37°C, gently flicking the tubes every 20 min for thorough mixture. Matrigel was added to the pellets after incubation and infected organoids were then seeded in 24-well cell culture plates and supplied with organoid growth medium. Infected organoids were kept in a humidified incubator with 5% (v/v) $CO_2$ at 35°C. Ct-infected mFGT organoids were collected on different days to undergo different analysis. (1) Organoid growth medium was collected from unbroken organoids and broken organoids, then analyzed by ELISA to test secreted ISG15 level; (2) protein samples of organoids were harvested for Immunoblotting.

## Transwell co-culture with mFGT organoids and NK cells

mFGT organoids with or without Ct infection were seeded in 12-well plates (Corning) as the lower chamber, each well included 3 individual 50 μl Matrigel drops with 1.3 ml organoid growth medium. Then the transwell membrane inserts (0.4 μm, Sigma Aldrich, Cat# PICM01250) were installed onto these wells, grow organoids in an incubator for 5 days. On the 5$^{th}$ day, 300 μl of freshly isolated NK cells (at 4 x 10$^6$ per ml) were seeded onto the transwell membrane as the upper chamber of this co-culture system. At the same time, Ct EBs, IL-12 (5 ng/ml), and/or ISG15 (25 ng/ml) according to designed experiments were added into the low chamber, and cultured for 36 h. Cell culture supernatants were collected from the upper transwell chamber to detect IFN-γ secretion levels via ELISA.

## Culture of bone marrow-derived macrophages (BMDMs) from mice

Harvest the femur and tibiae from mice and remove the muscle tissues adjoining the bone. Cut off the ends of the femur and tibia to remove the epiphyses, exposing the marrow. Fill a 2 mL syringe with a 23G needle with RPMI medium + 10% FBS, insert the syringe needle into one end of the bone, and flush out the bone marrow. Use a 1 mL pipette to pipette up and down to break cell clumps. Centrifuge the bone marrow suspension at 500 × g for 5 min at 4°C. Re-suspend the cell pellet in 2 mL of ACK lysis buffer and incubate on ice for 3 mins to eliminate RBCs. Centrifuge the bone marrow suspension again at 500 × g for 5 min at 4°C. Re-suspend the cell pellet gently in 1 mL of RPMI medium + 10% FBS, then filter the cell suspension through a 70 μm cell strainer to remove dead cell clumps and debris. Seed the isolated bone marrow cells at the desired concentration and culture them in the conditioned medium (RPMI medium + 10% FBS, + 10 ng/mL M-CSF) in a cell culture incubator with 37°C, 5% v/v CO2 for 7 days to differentiate into BMDMs. On day 4 post-extraction, add an equal volume of fresh conditioned medium to the culture.

### *In vitro* stimulations and polarization analysis of BMDMs via FACS

On day 7 post-extraction, BMDMs were treated with different stimuli. The culture medium of BMDMs was discarded and replaced with 2 mL of RPMI/10% FBS medium containing either 40 ng/mL IFN-γ, 40 ng/mL IL-4, 40 ng/mL IL-10, 40 ng/mL ISG15, or left unstimulated (as the medium control). Cells were incubated for 28 hours for further FACS analysis.

For FACS staining, the cell culture medium was discarded and BMDMs were harvested using a cell scraper (Sarstedt, Cat# 83.3950). After one washing step in DPBS, cells were pelleted at 500xg for 5min at 4°C. The cells were resuspended in Zombie NIR (1:1000, BioLegend, Cat# 423105) in PBS for live/dead staining and incubated for 10 min at room temperature. After an additional washing step, cells were incubated with Fc block antibody (1:200, BioLegend, Cat# 101302) in PBS for 15–20 minutes on ice. The cells were pelleted and washed twice with FACS buffer (BioLegend, Cat# 420201) before they were resuspended in antibody reaction buffer and incubated for 30 minutes on ice in the dark. Cells were again washed twice with FACS buffer and resuspended in 200 μL of FACS buffer.

Antibodies used: Anti-mouse F4/80 BV 605 (1:200, BioLegend, Cat# 123133), Anti-mouse CD11b-PE Cy7 (1:200, Thermo Fisher, Cat# 25-0112-82), Anti-mouse CD86-PE (1:200, Thermo Fisher, Cat# 12-0862-83), Anti-mouse CD163-BV711 (1:200, BioLegend, Cat# 155325), Anti-mouse CD206-APC (1:200, BioLegend, Cat# 141708). Cells were analyzed using a Thermo Attune NxT Flow Cytometer, and data were processed using FlowJo software.

### Western blotting and antibodies

To harvest protein samples, cells were washed with DPBS one time and lysed in 2x Laemmli buffer (10% 1.5 M Tris-HCl pH 6.8, 4% SDS, 30% glycerol and 1.5% ß-mercaptoethanol) on ice for 10 min incubation, and boiled at 95°C for 5min. Organoids were released from Matrigel using cold DPBS, centrifuged (1,000 g, 4°C for 5min) and lysed in 2x Laemmli buffer as well. Protein samples were separated by 10% SDS-PAGE gel and transferred onto a PVDF membrane (Sigma-Aldrich) by a semi-dry electroblotter (GENE MATE). Blots were blocked in 1% BSA/TBS-T (TBS-T, 1x Tris-buffered saline with 0.5% Tween-20) block buffer for 1 h at RT, then incubated with primary antibody in 1% BSA/TBS-T overnight at 4°C. The primary antibodies were used: anti-ISG15 Ab (1:400), anti-chlamydial HSP60 Ab (1:1,000) and anti-GAPDH Ab (1:1,000). Blots were incubated with corresponding HRP-conjugated secondary antibody (Santa Cruz) in 5% BSA/TBS-T block buffer for 1 h at RT. Proteins were then detected using the ECL system and Intas Chem HR 16–3200 reader.

### Mice and mouse Ct infection model

ISG15$^{-/-}$ mice (strain name: B6.129P2-isg15$^{tm1kpk}$/J [71]) were obtained from Klaus-Peter Knobeloch (Institute of Neuropathology, University of Freiburg, Germany). C57BL/6J mice were set as experimental wild-type (WT) controls. NKp46$^{iCre}$ (B6(Cg)-*Ncr1*$^{tm1.1(icre)Viv}$/Orl) mice were previously generated and kindly provided by E. Vivier [72] and crossed to IL18Ra$^{fl/fl}$ mice. The IL18Ra$^{fl/fl}$ line was generated and provided by Wolfgang Kastenmüller.

We used the trans-cervical inoculation method via the non-surgical embryo transfer device (NSET) (ParaTechs) to establish female upper genital tract infection with *Chlamydia trachomatis* L2/434/Bu strain (Ct) in mice. Five days prior to infection, 6–7 weeks-old female WT mice and ISG15$^{-/-}$ mice were injected subcutaneously with 2.5 mg each of Medroxyprogesterone acetate (MPA, Sigma Aldrich, Cat# M0250000) to eliminate the estrous cycle variable and enhance infection rate. Mice were challenged with infectious doses 2.5 x 10$^7$ IFUs of Ct EBs via NSET; mock-challenged mice received sterile SPG buffer. At defined time point after infection peripheral blood samples were obtained for analysis of cytokine levels in serum. Then mice were sacrificed, female genital tract (FGT) tissues were collected to undergo further different analysis, including the tissue immunofluorescence staining, the tissue FACS assay, as well as Ct burden detection and cytokine detection on the homogenized FGT tissues.

## Tissue immunofluorescence staining and antibodies

Mice were sacrificed at time points of interest after infection, intact mFGT tissues were collected and placed into 2 ml of BD Cytofix/Cytoperm buffer (BD Biosciences, Cat# 554714)/PBS (1:4) solution, overnight at 4°C for fixation and permeabilization. Tissues were washed with PBS 3 times and further dehydrated in 2 ml of 30% sucrose in PBS solution with 0.01% sodium azide overnight at 4°C. Processed tissues were embedded in cryostat molds filled with Tissue-Tek O.C.T. compound (Hartenstein, Cat# TTEK) and then frozen on dry ice. Serial horizontal sections were prepared via cryosection and mounted on silane-coated slides (VWR International). For histology, a blocking buffer containing 0.1 M Tris (Roth, Cat# 9090.2), 1% gelatin from cold water fish skin (Sigma-Aldrich, Cat. #G7041), 1% BSA (Roth, Cat # 8076.3), 0.3% Triton-X-100 (Roth, Cat# 3051.2), 2% normal mouse serum (Thermo Fisher, Cat# 10410), 5% horse serum (Sigma Aldrich, Cat# H0146-10ML) and 0.01% sodium azide was used. Primary staining as performed overnight at 4°C and secondary staining for 1-2h, also at 4°C. Slides were mounted using Fluoromount G (Thermo Fisher, Cat# 00-4958-02). Images were acquired on a Leica SP8 microscope and processed using Imaris software.

Antibodies used: CD16/32 (BioXcell, Cat# BE0307, 1:200), CD3 AF700 (Biolegend, Cat# 100215, 1:150), CD31 BV480 (BD, Cat# 565629, 1:200), E-cadherin AF488 (eBioscience, Cat# 53-3249-80, 1:100), goat IgG secondary antibody AF568 (Invitrogen, Cat# A11057, 1:1000), Ly6G APC (Biolegend, Cat# 127614, 1:250), MHC-II Pacific Blue (Biolegend, Cat# 107620, 1:300), Kp46 unconjugated (R&D systems, Cat# AF2225, 1:250).

## *Chlamydia* loads in tissues

To quantitate live *Chlamydia* in mFGT tissue, mice infected with Ct or mock-infected controls were sacrificed according to experimental plan. Intact mFGT tissues were isolated and transferred into a sterile 2 ml Eppendorf tube containing 1 ml SPG buffer and 6–8 glass beads. Tissues were homogenized at 4°C. The homogenates were briefly sonicated and centrifuged at 5,000 rpm for 10 min to pellet debris. Ct released into the supernatants were titrated on pre-prepared McCoy cells (70% confluency) [73]. After 24–28 h inoculation, inclusions were counted in five random fields per well under a microscope. The IFUs per tissue were calculated based on the mean number of IFUs per well and the dilution ratio. The IFUs per tissue were expressed as $\log_{10}$ IFUs.

## Quantification of cytokines by ELISAs

Culture medium from organoids and cells after stimulation or infection, as well as lavage fluid from the FGT in mice, were centrifuged at 1,000 × g for 10 min at 4°C to remove dead cells and debris. Supernatants were then collected. Serum from peripheral blood in mice was also collected. Cytokine levels in supernatants and serum samples were quantified via ELISA kits, according to manufacturer's protocols. OD values were measured using a TECAN M Plex microplate reader. In this study, we used human ISG15 ELISA Kit (antibodies, Cat# ABIN6957216), mouse ISG15 ELISA Kit (antibodies, Cat# ABIN6973820), mouse Interferon-gamma (IFN-γ) ELISA Kit (Abcam, Cat# ab100689), mouse Interleukin 10 (IL-10) ELISA Kit (Abcam, Cat# ab255729), mouse Interleukin 6 (IL-6) ELISA Kit (Invitrogen, Cat# 88-7064-22), mouse TNF-α ELISA Kit (Invitrogen, Cat# 88-7324-22).

## Statistical analyses

Except for *Chlamydia* load analyses, which used $\text{Log}_{10}$ transformed data, all statistical analyses were directly performed using GraphPad Prism 9.0 Software. Statistical analyses were performed with three biological replicates at least. Data are presented as mean ± SEM. Pearson's correlation coefficient was used to examine correlations.

## Ethics statement

Mouse studies were carried out in strict accordance with the guidelines for animal care and experimentation of German Animal Protection Law approved under the Animal (Scientific Procedures) Act 1986 (project license is TV-AZ 55.2.2-2532-2-762, Regierung von Unterfranken). All mice were euthanized at detection time points by cervical dislocation.

Primary ectocervix epithelial cells (EcCxEp) were isolated from human ectocervical biopsies provided by the Gynaecology Department of the University Hospital, Wuerzburg. The ethical vote for the scientific research use was approved by University of Wuerzburg ethics committee (Approval 261/20) and informed written consent was obtained from all donors.

## Supporting information

**S1 Fig. a. Bright-field microscopy images of Ct inclusions in mFGT organoids at 2 dpi.** The release pattern of Ct inclusions in the mFGT organoids was observed as apical release into the organoid lumen (pink arrows) as well as basolateral release into the extracellular matrix (blue arrows), primarily through apical release. b. ISG15 expression in mFGT organoids infected with Cm for 5 days. ISG15, HSP60, and GAPDH expression of infected and non-infected mFGT organoids were analyzed by Western blotting. A representative blot from three independent experiments is shown.
(TIFF)

**S2 Fig. Analysis of the purity of enriched NK cells by FACS assay.** Primary NK cells were enriched from mouse splenocytes using negative magnetic selection, and the purity of enriched NK was verified via FACS using anti-NK1.1-FITC antibody and anti-CD3-APC antibody.
(TIFF)

**S3 Fig. a-b. ISG15 also increases IFN-γ production by memory-phenotype CD8+ T cells.** For experimental and statistical details see Fig 2c and 2d. Shown here are whole splenocyte cultures, gated on memory-phenotype CD8+ T cells gated as CD3+ CD8a+ CD44$^{hi}$.
(TIFF)

**S4 Fig. Immunofluorescence staining of the mouse FGT at 12 hours post-infection to observe immune cells.** Wild-type mice, either infected with Ct or mock-infected with PBS, were sacrificed at 12 hpi. The entire genital tracts were then isolated and processed into cryosections for immunostaining. The presence of neutrophils (Ly6G), T cells (CD3) and NK cells (NKp46) was analyzed alongside MHC-II expression and the structural markers CD31 and E-cadherin.
(TIFF)

**S5 Fig. a. Representative FACS plots showing the gating strategy of characterizing CD86+, CD206+, and CD163+ cells in the BMDMs population (live CD11b+, F4/80+).** b. The frequency of CD86+ cells and the MFI of CD86 in gated ISG15$^{-/-}$ Mφ. ISG15$^{-/-}$ Mφ stimulated as described in Fig 6b, MFI is presented as Geomean values, data were analyzed using one-way ANOVA, n = 5. c. The frequency of CD206+ cells and the MFI of CD206 in gated ISG15$^{-/-}$ Mφ stimulated as described in Fig 6b. Data were analyzed using one-way ANOVA, n = 5. (Statistical significance is indicated as follows: *$p \leq 0.05$; ***$p \leq 0.001$; ****$p \leq 0.0001$; ns, not statistically significant; n indicates independent biological replicates).
(TIFF)

**S1 Data. ISG15 secretion levels in the supernatants of different cell lines, as detected by ELISA.**
(XLSX)

## Acknowledgments

We thank Dr. Yongzheng Wu and Dr. Agathe Subtil for scientific exchange on ISG15 prior to publication. We thank Dr. Eric Vivier and Dr. Wolfgang Kastenmüller for kindly provided NKp46$^{iCre}$ (B6(Cg)-*Ncr1*$^{tm1.1(icre)Viv}$/Orl) mice and IL18Ra$^{fl/fl}$ mice. We thank Dr. Jörg Wischhusen for the human Fimb cells. We thank Dr. Qi Tian for his support in establishing tissue immunofluorescence staining. We thank Dr. Andreas Demuth for support with the animal experiment application. We thank Dr. Naveen C.R and Dr. Sophie Hügelschäffer for advice and support with experiments and Andrea Fick and Sarah Hess for technical support in animal genotyping and care.

## Author contributions

**Conceptualization:** Yongxia Guo.

**Formal analysis:** Yongxia Guo.

**Funding acquisition:** Georg Gasteiger, Thomas Rudel.

**Investigation:** Yongxia Guo, Sigrun V. Stulz, David Komla Kessie, Nadine Vollmuth, Tommaso Torcellan, Klaus-Peter Knobeloch.

**Methodology:** Yongxia Guo, Sigrun V. Stulz.

**Resources:** Georg Gasteiger, Thomas Rudel.

**Supervision:** Georg Gasteiger, Thomas Rudel.

**Visualization:** Yongxia Guo.

**Writing – original draft:** Yongxia Guo.

**Writing – review & editing:** Yongxia Guo, Sigrun V. Stulz, Georg Gasteiger, Thomas Rudel.

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
