## [Decision Letter · Decision Letter 0]

PPATHOGENS-D-25-00470

Secreted ISG15 induced by Chlamydia trachomatis infection exerts immunomodulatory effects on IFN-γ defense and inflammation

PLOS Pathogens

Dear Dr. Rudel,

Thank you for submitting your manuscript to PLOS Pathogens. After careful consideration, we feel that it has merit but does not fully meet PLOS Pathogens's publication criteria as it currently stands. Therefore, we invite you to submit a revised version of the manuscript that addresses the points raised during the review process.

Please submit your revised manuscript within 60 days May 20 2025 11:59PM. If you will need more time than this to complete your revisions, please reply to this message or contact the journal office at plospathogens@plos.org. Please include the following items when submitting your revised manuscript:

We look forward to receiving your revised manuscript.

Kind regards,

Stephen J McSorley

Guest Editor

PLOS Pathogens

Matthew Wolfgang

Section Editor

PLOS Pathogens

 Sumita Bhaduri-McIntosh

Editor-in-Chief

PLOS Pathogens

orcid.org/0000-0003-2946-9497

 Michael Malim

Editor-in-Chief

PLOS Pathogens

orcid.org/0000-0002-7699-2064

**Additional Editor Comments:**

Authors should address all reviewer comments but pay particular attention to the issues raised by Reviewer 2 with regard to the data that conflict with the model.

**Journal Requirements:**

At this stage, the following Authors/Authors require contributions: Yongxia Guo, Sigrun V. Stulz, David Kessie, Tommaso Torcellan, Nadine Vollmuth, Qi Tian, Klaus-Peter Knobeloch, Georg Gasteiger, and Thomas Rudel. Please ensure that the full contributions of each author are acknowledged in the "Add/Edit/Remove Authors" section of our submission form.

https://journals.plos.org/plospathogens/s/submission-guidelines#loc-parts-of-a-submission

3) We noticed that you used the phrase 'data not shown' in the manuscript. We do not allow these references, as the PLOS data access policy requires that all data be either published with the manuscript or made available in a publicly accessible database. Please amend the supplementary material to include the referenced data or remove the references.

4) We do not publish any copyright or trademark symbols that usually accompany proprietary names, eg ©,  ®, or TM  (e.g. next to drug or reagent names). Therefore please remove all instances of trademark/copyright symbols throughout the text, including:

- ® on pages: 22 and 23.

- TM on pages: 32 and 47.

5) Please upload all main figures as separate Figure files in .tif or .eps format. For more information about how to convert and format your figure files please see our guidelines: 

Potential Copyright Issues:

- Figures 1, 4, 5, and 7. Please confirm whether you drew the images / clip-art within the figure panels by hand. If you did not draw the images, please provide a link to the source of the images or icons and their license / terms of use; or written permission from the copyright holder to publish the images or icons under our CC BY 4.0 license. Alternatively, you may replace the images with open source alternatives. See these open source resources you may use to replace images / clip-art:

7) Please ensure that the funders and grant numbers match between the Financial Disclosure field and the Funding Information tab in your submission form. Note that the funders must be provided in the same order in both places as well. State what role the funders took in the study. If the funders had no role in your study, please state: "The funders had no role in study design, data collection and analysis, decision to publish, or preparation of the manuscript.".

**Reviewers' Comments:**

Reviewer's Responses to Questions

**Part I - Summary**

Reviewer #1: The authors performed in vitro and in vivo analyses to examine the role of ISG15 during Chlamydia infection. These studies revealed that murine Ct-infection induced ISG15 production from primary epithelial cells that enhanced NK cell interferon-gamma expression. This inteferon-gamma signaling effect required IL-18 activation of NK cells. The lack of ISG15 in Ct-infected mice modestly reduced bacterial burden and significantly enhanced uterine horn pathology, which was associated with lower interferon-gamma and IL-10 production, respectively. Overall, this is a novel study with well designed experiments and appropriate statistical analyses. The conclusions are well-supported by the data and appropriate citations are used. Overall, the manuscript is of very good merit. To be acceptable for publication, a few minor corrections should be addressed and a couple of additional experiments should be performed that would substantially increase the quality of the manuscript and make for a more comprehensive study.

Reviewer #2: In this study, the authors find that Chlamydia trachomatis (CT) infection robustly induces the expression and extracellular secretion of ISG15 locally in vitro and in vivo. They report that ISG15 acts as a critical immunomodulatory cytokine during CT infection. Specifically, they found IL-18, in addition to IL-12, as a novel co-stimulatory cytokine, that acts in synergy with ISG15 to strongly enhance IFNγ production at early stages of. Whereas at later stages, ISG15 promotes IL-10 production to prevent excess inflammatory response and tissue damage. Overall, the experiments are well designed and convincing. However, some of the data seem to contradict the model and correlations are too easily interpreted as causations. While this can be experimentally approached, the authors may alternatively consider to tone down some of their conclusions and adjust the working model.

Reviewer #3: This study provides compelling evidence showing the induction of ISG15 in the FGT during Chlamydia infection and the

immunomodulatory role of ISG15 in innate immune responses in vitro. Data suggest that secreted ISG15 in the FGT increases IFNg release from NK cells in a IL18-dependent manner. Transcervical Ct infection in ISG15-deficient mice resulted in reduced IFNg at early stage of infection (3 dpi), and increased pathology and bacterial burden at later stage (14 dpi) which correlate with decreased IL-10. This study provides new insights into the critical dual function of ISG15 in modulating host inflammatory responses during Chlamydia infection.

**Part II – Major Issues: Key Experiments Required for Acceptance**

Reviewer #1: Was C. muridarum used in any parallel experiments specifically in mouse cells to confirm ISG15 production in a strain-independent manner using the host species? This would further enhance the quality of the manuscript since it would have further implications for testing in mouse models. Additional in vitro experiments, as performed in Fig 1D-E, are warranted to demonstrate and compare ISG15 production after Cm and Ct infection, at minimum.

To validate in vitro findings from Figure 6, an ex vivo flow cytometric analysis of CD86/CD163/CD206 expression on macrophages in WT and ISG15-/- mice after infection (e.g. day 3 and 14) is warranted. A co-analysis of marker expression on monocytes (and PMNs, if possible) would further improve the study.

Reviewer #2: 1. The paper shows that significant differences in IFNγ production can be detected between WT and ISG15 KO mFGT at 3dpi (Figure 4d). However, no difference in bacterial burden is observed until 14dpi (Figure 4c) when IFNγ levels are no longer different between ISG15 WT and KO mFGT (Figure 4d). These data seem to contradict the proposed model that diminished IFNγ production in ISG15 KO mice is responsible for the elevated CT burden in ISG15 KO mice. To prove causality IFNγ blocking antibodies could be used. The expected outcome would be that the difference in CT burden between WT and ISG15 KO mice would disappear once IFNγ signaling is blocked. Alternatively, the authors should acknowledge that the data actually contradicts their model and come up with a new model to explain the ISG15 KO burden phenotype.

2. The authors propose that ISG15 signals to nearby NK cells in mFGT to enhance the IFNγ response during CT infection. In support of this model the paper provides an image that possibly shows NK cells near epithelial cells in WT mFGT at 12hpi versus mock-infected mFGT (Figure S4). However, it is not clear why 12hpi was chosen as a timepoint and whether NK cells were also present in ISG15 KO mFGT. The authors should consider not showing Figure S4, as these data lack controls and are not quantified. Maybe a better approach would be to perform FACS analyses on immune cells present at WT versus ISG15 KO mFGT during C. trachomatis infection to strengthen their findings of NK cell recruitment/infiltrate to infected the epithelium

3. The authors should include all necessary controls (untreated/uninfected control, IFNγ alone, ISG15 alone, and C. trachomatis infection alone) when assessing cytokine secretions (Figure 6 f-h).

Reviewer #3: 1. The authors show in Fig. 2e that ISG15 regulates NK cell IFNg production in a cell-extrinsic manner. However, in the organoids culture system in Fig. 3b, it appears that ISG15 secreted from the organoids directly stimulates NK cells, as no other cell type exist in this system. Please explain the discrepancy.

2. The macrophage data in Fig. 5 and 6 relies heavily on CD163 as a M2 marker. However, data in Fig. 5b does not support CD163 as a good murine M2 marker (as opposed to human macrophage). The positive control IL-4 does not induce a prominent CD163+ population. In contrast, CD206 works better as a M2 marker, but is not pursued. There is also a lack of functional readouts such as iNOS and Arg in support of M1/M2 phenotype.

3. In vivo phenotyping of IFNg+ NK cell and M1/M2 macrophages in WT and ISG15-/- mice would significantly strengthen the study.

**Part III – Minor Issues: Editorial and Data Presentation Modifications**

Reviewer #1: Abstract - “ISG15 can exist as a covalently conjugated protein or as an unconjugated cytokine.” Please revise to something like ‘can exist in unconjugated form or conjugated via ISGylation to other host proteins’ to better detail its function.

Line 72 – change ‘aiming’ to ‘aimed’.

Line 87-89 - The writing could be better parsed regarding the findings from citation 26 to more clearly state interferon-gamma redundancy occurs specifically in the context of dissemination.

Line 234 – It states that hydrosalpinx was evaluated. Ct causes minimal oviduct disease in mice and is usually analyzed significantly past day 14 post-infection. Uterine horn pathology was assessed but oviduct scores are absent. If no hydrosalpinx/oviduct dilation was observed, then this should be stated.

Line 286 – Please rephrase to indicate that citations 58,59 are from other disease models. The writing could be misconstrued that it is from a Chlamydia model.

Line 307 – Like above, please specify if observations are made in Chlamydia or other pathogen models. Please review the entire manuscript to better clarify these instances.

Figure 6 - The IFNg label on the x-axis is cut off in panels b-d.

Lines 664-669. The methods suggest macrophages were analyzed ex vivo after murine infection. If so, this would help address the above major critique regarding analyzing macrophage marker expression in mice post-infection.

Reviewer #2: 1. One key notion the authors were trying to demonstrate was the low level of IL-18 in ISG15 stimulation of IFNγ (Figure 2d-e). It would provide more solid ground for this hypothesis if IL-18 secretions during infections were assessed.

2. The authors showed that intracellular ISG15 is dispensable for the C. trachomatis infection, and the conjugation of ISG15 is not the focus of the study. However, showing and assessing the free ISG15 levels in all SDS-PAGE analyses alone are not sufficient to rule out changes and involvements of ISG15-conjugation. The authors should consider providing full blots containing the conjugated upper bands in the supplemental figure to further support their findings.

3. The authors mentioned that basal level of gamma between WT and ISG15 KO (line 246-247) without citing any previous study. Was this data not shown or previously established and should be cited?

4. The authors proposed that at later time of the infection, ISG15 acts through IL-10 to modulate macrophage functions for tissue repair. However, the authors only assessed IL-10 and IFNγ levels (Figure 4d, 4g-i) and not other immunomodulatory cytokines. Supplementing IL-10 exogenously and monitor the effect on animal pathology in ISG15 KO vs WT mice would provide evidence for a causal link. Without such data, the authors should acknowledge the limitations of their correlative studies

5. Lastly, the authors demonstrated that ISG15 stimulation of macrophages in vitro leads to the polarization of M2-like macrophages (7 days of differentiation followed by 24h cytokine stimulation). They propose these M2-like macrophages (CD163hi) account for the tissue damage control at late infection. The authors may consider using FACS analyses to demonstrate macrophages present in the WT but not ISG15 KO mFGT to have similar signatures particularly at 7, 14, and 28 dpi.

Reviewer #3: (No Response)

PLOS authors have the option to publish the peer review history of their article (what does this mean? ). If published, this will include your full peer review and any attached files.

**Do you want your identity to be public for this peer review?** For information about this choice, including consent withdrawal, please see our Privacy Policy .

Reviewer #1: No

Reviewer #2: No

Reviewer #3: No

**Figure resubmission:**
---

## [Decision Letter · Decision Letter 1]

Dear Dr. Rudel,

We are pleased to inform you that your manuscript 'Secreted ISG15 induced by Chlamydia trachomatis infection exerts immunomodulatory effects on IFN-γ defense and inflammation' has been provisionally accepted for publication in PLOS Pathogens.

Best regards,

Stephen J McSorley

Guest Editor

PLOS Pathogens

Matthew Wolfgang

Section Editor

PLOS Pathogens

Sumita Bhaduri-McIntosh

Editor-in-Chief

PLOS Pathogens

orcid.org/0000-0003-2946-9497

Michael Malim

Editor-in-Chief

PLOS Pathogens

orcid.org/0000-0002-7699-2064

Reviewer Comments (if any, and for reference):

Reviewer's Responses to Questions

**Part I - Summary**

Reviewer #1: The authors have fully addressed the points raised by reviewers.

Reviewer #2: my concerns were sufficiently addressed in the revised MS

Reviewer #3: My concerns were sufficiently addressed.

**Part II – Major Issues: Key Experiments Required for Acceptance**

Reviewer #1: All points were satisfactorily addressed.

Reviewer #2: (No Response)

Reviewer #3: (No Response)

**Part III – Minor Issues: Editorial and Data Presentation Modifications**

Reviewer #1: All points were addressed.

Reviewer #2: (No Response)

Reviewer #3: (No Response)

PLOS authors have the option to publish the peer review history of their article (what does this mean? ). If published, this will include your full peer review and any attached files.

**Do you want your identity to be public for this peer review?** For information about this choice, including consent withdrawal, please see our Privacy Policy .

Reviewer #1: No

Reviewer #2: No

Reviewer #3: No

---

## [Editor Report · Acceptance letter]

Dear Dr. Rudel,

We are delighted to inform you that your manuscript, "Secreted ISG15 induced by *Chlamydia trachomatis* infection exerts immunomodulatory effects on IFN-γ defense and inflammation," has been formally accepted for publication in PLOS Pathogens.

Best regards,

Sumita Bhaduri-McIntosh

Editor-in-Chief

PLOS Pathogens

orcid.org/0000-0003-2946-9497

Michael Malim

Editor-in-Chief

PLOS Pathogens

orcid.org/0000-0002-7699-2064